# PeFLL: Personalized Federated Learning by Learning to Learn

**Jonathan Scott**
Institute of Science and Technology Austria (ISTA)
jonathan.scott@ist.ac.at

**Hossein Zakerinia**
Institute of Science and Technology Austria (ISTA)
hossein.zakerinia@ist.ac.at

**Christoph H. Lampert**
Institute of Science and Technology Austria (ISTA)
chl@ist.ac.at

## Abstract

We present PeFLL, a new personalized federated learning algorithm that improves over the state-of-the-art in three aspects: 1) it produces more accurate models, especially in the low-data regime, and not only for clients present during its training phase, but also for any that may emerge in the future; 2) it reduces the amount of on-client computation and client-server communication by providing future clients with ready-to-use personalized models that require no additional finetuning or optimization; 3) it comes with theoretical guarantees that establish generalization from the observed clients to future ones.

At the core of PeFLL lies a *learning-to-learn* approach that jointly trains an embedding network and a hypernetwork. The embedding network is used to represent clients in a latent descriptor space in a way that reflects their similarity to each other. The hypernetwork takes as input such descriptors and outputs the parameters of fully personalized client models. In combination, both networks constitute a learning algorithm that achieves state-of-the-art performance in several personalized federated learning benchmarks.

## 1 Introduction

Federated learning (FL) (McMahan et al., 2017) has emerged as a de-facto standard for privacy-preserving machine learning in a distributed setting, where a multitude of clients, e.g., user devices, collaboratively learn prediction models without sharing their data directly with any other party. In practice, the clients' data distributions may differ significantly from each other, which makes learning a single global model sub-optimal (Li et al., 2020; Kairouz et al., 2021). Personalized federated learning (pFL) (Smith et al., 2017) is a means of dealing with such statistical heterogeneity, by allowing clients to learn individual models while still benefiting from each other.

A key challenge of personalized federated learning lies in balancing the benefit of increased data that is available for joint training with the need to remain faithful to each client's own distribution. Most methods achieve this through a combination of global model training and some form of finetuning or training of a client-specific model. However, such an approach has a number of shortcomings. Consider, for instance, a federated learning system over millions of mobile devices, such as *Gboard* (Hard et al., 2018). Only a fraction of all possible devices are likely to be seen during training, devices can drop out or in at any time, and it should be possible to produce models also for devices that enter the system at a later time. Having to train or finetune a new model for each such new device incurs computational costs. This is particularly unfortunate when the computation happens on the client devices, which are typically of low computational power. Furthermore, high communication costs emerge when model parameters or updates have to be communicated repeatedly between the server and the client. The summary effect is a high latency before the personalized model is ready for use. Additionally, the quality of the personalized model is unreliable, as it depends on the amount of data available on the client, which in personalized federated learning tends to be small.

In this work, we address all of the above challenges by introducing a new personalized federated learning algorithm. In PeFLL (for *Personalized Federated Learning by Learning to Learn*), it is the server that produces ready-to-use personalized models for any client by means of a single forward pass through a *hypernetwork*. The hypernetwork's input is a descriptor vector, which any client can produce by forward passing some of its data through an *embedding network*. The output of the hypernetwork is the parameters of a personalized model, which the client can use directly without the need for further training or finetuning. Thereby **PeFLL overcomes the problems of high latency and client-side computation cost** in model personalization from which previous approaches suffered when they required client-side optimization and/or multi-round client-server communication.

The combined effect of evaluating first the embedding network and then the hypernetwork constitutes a *learning method*: it produces model parameters from input data. The process is parametrized by the networks' weights, which PeFLL trains end-to-end across a training set of clients. This *learning-to-learn* viewpoint allows for a number of advantageous design choices. For example, **PeFLL benefits when many clients are available for training**, where other approaches might suffer in prediction quality. The reason is that the client models are created by end-to-end trained networks, and the more clients participate in the training and the more diverse these are, the better the networks' generalization abilities to future clients. Also, **PeFLL is able to handle clients with small datasets** better than previous approaches. Because the descriptor evaluation is a feed-forward process, it is not susceptible to overfitting, as any form of on-client training or finetuning would be. At the same time, already during training the client descriptors are computed from mini-batches, thereby anticipating the setting of clients with little data. Once the training phase is concluded, **PeFLL can even be used to create personalized models for clients with only unlabeled data.**

PeFLL's strong empirical results, on which we report in Section 4 are not a coincidence, but they can be explained by its roots in theory. Its training objective is derived from a **new generalization bound**, which we prove in Section 3.2 and which yields tighter bounds in the federated setting than previous ones. PeFLL's training procedure distributes the necessary computation between the server and the clients in a way that ensures that the computationally most expensive task happen on the server yet stays truthful to the federated paradigm. The optimization is stateless, meaning that clients can drop out of the training set or enter it at any time. Nevertheless, as we show in Section 3.1, it **converges at a rate comparable to standard federated averaging**.

## 2 RELATED WORK

*Personalized federated learning (pFL)* is currently one of the promising techniques for learning models that are adapted to individual clients' data distributions without jeopardizing their privacy. Existing approaches typically follow one of multiple blueprints: *multi-task methods* (Smith et al., 2017; Marfoq et al., 2021; Dinh et al., 2022; Li et al., 2021; Dinh et al., 2020; Hanzely et al., 2020; Hanzely and Richtárik, 2020; Ye et al., 2023; Zhang et al., 2023; Qin et al., 2023) learn individual per-client models, while sharing information between clients, e.g. through regularization towards a central model. *Meta-learning methods* learn a shared model, which the clients can quickly adapt or finetune to their individual data distributions, e.g. by a small number of gradient updates (Fallah et al., 2020; Jiang et al., 2019; Wu et al., 2023a). *Decomposition-based methods* (Arivazhagan et al., 2019; Collins et al., 2021; Bui et al., 2019; Liang et al., 2020; Chen et al., 2023; Wu et al., 2023b) split the learnable parameters into two groups: those that are meant to be shared between clients (e.g. a feature representation) and those that are learned on a per-client basis (e.g. classification heads). *Clustering-based methods* (Ghosh et al., 2020; Mansour et al., 2020) divide the clients into a fixed number of subgroups and learn individual models for each cluster.

*Hypernetworks* (Ha et al., 2017) have only recently been employed in the context of pFL (Ma et al., 2022; Yi et al., 2023). Typically, the hypernetworks reside on the clients, which limits the methods' efficiency. Closest to our work are Shamsian et al. (2021) and Li et al. (2023), which also generate each clients' personalized model using server-side hypernetworks. These works adopt a non-parametric approach in which the server learns and stores individual descriptor vectors per client. However, such an approach has a number of downsides. First, it entails a stateful optimization problem, which is undesirable because it means the server has to know at any time in advance which client is participating in training to retrieve their descriptors. Second, the number of parameters stored at the server grows with the number of clients, which can cause scalability issues in large-scale applications. Finally, in such a setting the hypernetwork can only be evaluated for clients with which

the server has interactive before. For new clients, descriptors must first to be inferred, and this requires an optimization procedure with multiple client-server communication rounds. In contrast, in PeFLL the server learns just the embedding network, and the client descriptors are computed on-the-fly using just forward-evaluations on the clients. As a consequence, clients can drop in or out at any time, any number of clients can be handled with a fixed memory budget, and even for previously unseen clients, no iterative optimization is required.

*Learning-to-Learn.* Outside of federated learning, the idea that learning from several tasks should make it easier to learn future tasks has appeared under different names, such as *continual learning* (Ring, 1994; Mitchell et al., 2015), *lifelong learning* (Thrun and Mitchell, 1995; Chen and Liu, 2018; Parisi et al., 2019), *learning to learn* (Thrun and Pratt, 1998; Andrychowicz et al., 2016), *inductive bias learning* (Baxter, 2000), *meta-learning* (Vilalta and Drissi, 2002; Finn et al., 2017; Hospedales et al., 2021). In this work, we use the term *learning-to-learn*, because it best describes the aspect that the central object that PeFLL learns is a mechanism for predicting model parameters from training data, i.e. essentially it learns a learning algorithm.

The central question of interest is if the learned learning algorithm *generalizes*, i.e., if it produces good models not only for the datasets used during its training phase, but also for future datasets. Corresponding results are typically generalization bounds in a PAC-Bayesian framework (Pentina and Lampert, 2014; Amit and Meir, 2018; Rothfuss et al., 2021; Guan and Lu, 2022; Rezazadeh, 2022), as we also provide in this work. Closest to our work in this regard is Rezazadeh (2022), which addresses the task of hyperparameter learning. However, their bound is not well adapted to the federated learning setting in which the number of clients is large and the amount of data per client might be small. For a more detailed comparison, see Section 3.2 and Appendix B.

## 3 METHOD

We now formally introduce the problem we address and introduce the proposed PeFLL algorithm. We assume a standard supervised federated learning setting with a (possibly very large) number, $n$, of clients. Each of these has a data set, $S_i = \{(x_1^i, y_1^i), \ldots, (x_{m^i}^i, y_{m^i}^i)\} \subset \mathcal{X} \times \mathcal{Y}$, for $i \in \{1, \ldots, n\}$, sampled from a client-dependent data distribution $D_i$. Throughout the paper, we adopt a *non-i.i.d.* setting, i.e., the data distributions can differ between clients, $D_i \neq D_j$ for $i \neq j$. For any model, $\theta \in \mathbb{R}^d$ (we identify models with their parameter vectors), the client can compute its training loss, $\mathcal{L}(\theta; S_i) = \frac{1}{m^i} \sum_{j=1}^{m^i} \ell(x_j^i, y_j^i, \theta)$, where $\ell : \mathcal{X} \times \mathcal{Y} \times \mathbb{R}^d \to \mathbb{R}_+$ is a loss function. The goal is to learn client-specific models, $\theta_i$, in a way that exploits the benefit of sharing information between clients, while adhering to the principles of federated learning.

PeFLL adopts a *hypernetwork* approach: for any client, a personalized model, $\theta \in \mathbb{R}^d$, is produced as the output of a shared deep network, $h : \mathbb{R}^l \to \mathbb{R}^d$, that takes as input a *client descriptor* $v \in \mathbb{R}^l$. To compute client descriptors, we use an embedding network, $\phi : \mathcal{X} \times \mathcal{Y} \to \mathbb{R}^l$, which takes individual data points as input and averages the resulting embeddings:[1] $v(S) = \frac{1}{|S|} \sum_{(x,y) \in S} \phi(x, y)$. We denote the hypernetwork's parameters as $\eta_h$ and the embedding network's parameters as $\eta_v$. As shorthand, we write $\eta = (\eta_h, \eta_v)$.

Evaluating the embedding network followed by the hypernetwork can be seen as a *learning algorithm*: it transforms a set of training data points into model parameters. Consequently, learning $\eta$ can be seen as a form of *learning-to-learn*. Specifically, we propose the *PeFLL (Personalized Federated Learning by Learning to Learn)* algorithm, which consists of solving the following optimization problem, where $\|\cdot\|$ denotes the $L^2$-norm,

$$\min_{\eta_h, \eta_v} \quad \lambda_h \|\eta_h\|^2 + \lambda_v \|\eta_v\|^2 + \sum_{i=1}^n \left[ \mathcal{L}\big( h(v(S_i; \eta_v); \eta_h); S_i \big) + \lambda_\theta \big\| h(v(S_i; \eta_v); \eta_h) \big\|^2 \right]. \quad (1)$$

PeFLL performs this optimization in a way that is compatible with the constraints imposed by the federated learning setup. Structurally, it splits the objective (1) into several parts: a *server objective*, which consist of the first two (regularization) terms, $f(\eta_h, \eta_v) = \lambda_h \|\eta_h\|^2 + \lambda_v \|\eta_v\|^2$, and multiple *per-client objectives*, each of which consists of the terms inside the summation $f_i(\theta_i; S_i) = \mathcal{L}(\theta_i; S_i) + \lambda_\theta \|\theta_i\|^2$. The terms are coupled through the identity $\theta_i = h(v(S_i; \eta_v); \eta_h)$. This split

---

[1]This construction is motivated by the fact that $v$ should be a permutation invariant (set) function (Zaheer et al., 2017), but for the sake of conciseness, one can also take it simply as an efficient design choice.

**Algorithm 1** `PeFLL-predict`

---

**input** target client with private dataset $S$
1: *Server sends embedding network $\eta_v$ to client*
2: *Client selects a data batch $B \subseteq S$*
3: *Client computes $v = v(B; \eta_v)$*
4: *Client sends descriptor $v$ to server*
5: *Server computes $\theta = h(v; \eta_h)$*
6: *Server sends personalized model $\theta$ to client*

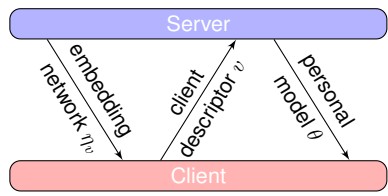

Figure 1: Communication protocol of PeFLL-predict for generating personalized models.

**Algorithm 2** `PeFLL-train`

---

**input** number of training steps, $T$
**input** number of clients to select per step, $c$
1: **for** $t = 1, \ldots, T$ **do**
2:     $P \leftarrow$ *Server randomly samples $c$ clients*
3:     *Server broadcasts embedding network $\eta_v$ to $P$*
4:     **for** client $i \in P$ in parallel **do**
5:         *Client selects a data batch $B \subseteq S_i$*
6:         $v_i \leftarrow v(B; \eta_v)$, *client computes descriptor*
7:         *Client sends $v_i$ to server*
8:         $\theta_i \leftarrow h(v_i; \eta_h)$ *server computes personalized model*
9:         *Server sends $\theta_i$ to client*
10:        $\theta_i^{\text{new}} \leftarrow$ client runs $k$ steps of local SGD on $f_i(\theta_i)$
11:        *Client sends $\Delta\theta_i := \theta_i^{\text{new}} - \theta_i$ to server*
12:       $(\Delta\eta_h^{(i)}, \Delta v_i) \leftarrow$ server runs `backprop` with error vector $\Delta\theta_i$
13:        *Server sends $\Delta v_i$ to client*
14:       $\Delta\eta_v^{(i)} \leftarrow$ client runs `backprop` with error vector $\Delta v_i$
15:        *Client sends $\Delta\eta_v^{(i)}$ to server*
16:     **end for**
17:     $\eta_h \leftarrow (1 - 2\beta\lambda_h)\eta_h + \frac{1}{|P|}\sum_{i \in P} \Delta\eta_h^{(i)}$
18:     $\eta_v \leftarrow (1 - 2\beta\lambda_v)\eta_v + \frac{1}{|P|}\sum_{i \in P} \Delta\eta_v^{(i)}$
19: **end for**
**output** network parameters $(\eta_h, \eta_v)$

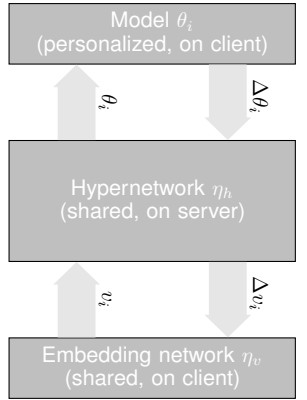

Figure 2: Data flow for PeFLL model generation (forward pass, left) and training (backward pass, right). The client descriptor, $v_i$, and the client model $\theta_i$ are small. Transmitting them and their update vectors is efficient. The hypernetwork, $\eta_h$, can be large, but it remains on the server.

allows PeFLL to distribute the necessary computation efficiently, preserving the privacy of the client data, and minimizing the necessary communication overhead. Pseudocode of the specific steps is provided in Algorithms 1 and 2.

We start by describing the `PeFLL-predict` routine (Algorithm 1), which can predict a personalized model for any target client. First, the server sends the current embedding network, $\eta_v$, to the client (line 1), which evaluates it on all or a subset of its data to compute the client descriptor (line 3). Next, the client sends its descriptor to the server (line 4), who evaluates the hypernetwork on it (line 5). The resulting personalized model is sent back to the client (line 6), where it is ready for use. Overall, only two server-to-client and one client-to-server communication steps are required before the client has obtained a functioning personalized model (see Figure 1).

The `PeFLL-train` routine (Algorithm 2) mostly adopts a standard stochastic optimization pattern in a federated setting, with the exception that certain parts of the forward (and backward) pass take place on the clients and others take place on the server. Thus PeFLL follows the Split Learning (Gupta and Raskar, 2018) paradigm in Federated Learning (Thapa et al., 2022). In each iteration the server selects a batch of available clients (line 2) and broadcasts the embedding model, $\eta_v$, to all of them (line 3). Then, each client in parallel computes its descriptor, $v_i$, (6), sends it to the server (line 7), and receives a personalized model from the server in return (line 8, 9). At this point the forward pass is over and backpropagation starts. To this end, each client performs local SGD for $k$ steps on its personalized model and personal data (line 10). It sends the resulting update, $\Delta\theta_i$, to the server (line 11), where it acts as a proxy for $\frac{\partial f_i}{\partial \theta_i}$. According to the chain rule, $\frac{\partial f_i}{\partial \eta_h} = \frac{\partial f_i}{\partial \theta_i} \frac{\partial \theta_i}{\partial \eta_h}$ and $\frac{\partial f_i}{\partial v_i} = \frac{\partial f_i}{\partial \theta_i} \frac{\partial \theta_i}{\partial v_i}$. The server can evaluate both expressions using backpropagation (line 12), because all

required expressions are available to it now. It obtains updates $\Delta\eta_h^{(i)}$ and $\Delta v_i$, the latter of which it sends to the client (line 13) as a proxy for $\frac{\partial f_i}{\partial v_i}$. Again based on the chain rule ($\frac{\partial f_i}{\partial \eta_v} = \frac{\partial f_i}{\partial v_i}\frac{\partial v_i}{\partial \eta_v}$), the client computes an update vector for the embedding network, $\Delta\eta_v^{(i)}$ (line 14), and sends it back to the server (line 15). Finally, the server updates all network parameters from the average of per-client contributions as well as the contributions from the server objective (lines 17, 18) [2].

The above analysis shows that PeFLL has a number of desirable properties: **1) It distributes the necessary computation in a client-friendly way**. The hypernetwork is stored on the server and evaluated only there. It is also never sent via the communication channel. This is important, because hypernetworks can be quite large, as already their output layer must have as many neurons as the total number of parameters of the client model. Therefore, they tend to have high memory and computational requirements, which client devices might not be able to provide.

**2) It has low latency and communication cost.** Generating a model for any client requires communicating only few and small quantities: a) the parameters of the embedding network, which are typically small, b) the client descriptors, which are low dimensional vectors, and c) the personalized models, which typically also can be kept small, because they only have to solve client-specific rather than general-purpose tasks. For the backward pass in the training phase, three additional quantities need to be communicated: a) the clients' suggested parameter updates, which are of equal size as the model parameters, b) gradients with respect to the client descriptors, which are of equal size as the descriptors, and c) the embedding network's updates, which are of the same size as the embedding network. All of these quantities are rather small compared to, e.g., the size of the hypernetwork, which PeFLL avoids sending.

**3) It respects the federated paradigm.** Clients are not required to share their training data but only their descriptors and—during training—the gradient of the loss with respect to the model parameters. Both of these are computed locally on the client devices. Note that we do not aim for formal privacy guarantees in this work. These could, in principle, be provided by incorporating *multi-party computation* (Evans et al., 2018) and/or *differential privacy* (Dwork and Roth, 2014). Since a descriptor is an average of a batch of embeddings, this step can be made differentially private with some added noise using standard DP techniques such as Gaussian or Laplace mechanisms. This, however, could add the need for additional trade-offs, so we leave the analysis to future work.

### 3.1 CONVERGENCE GUARANTEES

We now establish the convergence of PeFLL's training procedure. Specifically, we give guarantees in the form of bounding the expected average gradient norm, as is common for deep stochastic optimization algorithms. The proof and full formulation can be found in Appendix C.

**Theorem 3.1.** *Under standard smoothness and boundedness assumptions (see appendix), PeFLL's optimization after $T$ steps fulfills*

$$\frac{1}{T}\sum_{t=1}^{T}\mathbb{E}\left\|\nabla F(\eta_t)\right\|^2 \leq \frac{(F(\eta_0)-F_*)}{\sqrt{cT}} + \frac{L(6\sigma_1^2+4k\gamma_G^2)}{k\sqrt{cT}} + \frac{224cL_1^2b_1^2b_2^2}{T} + \frac{8b_1^2\sigma_2^2}{b} + \frac{14L_1^2b_2^2\sigma_3^2}{b}, \quad (2)$$

*where $F$ is the PeFLL objective* (1) *with lower bound $F_*$. $\eta_0$ are the parameter values at initialization, $\eta_1, \ldots, \eta_T$ are the intermediate parameter values. $L, L_1$ are smoothness parameters of $F$ and the local models. $b_1, b_2$ are bounds on the norms of the gradients of the local model and the hypernetwork, respectively. $\sigma_1$ is a bound on the variance of stochastic gradients of local models, and $\sigma_2, \sigma_3$ are bounds on the variance due to the clients generating models with data batches of size $b$ instead of their whole training set. $\gamma_G$ is a bound on the dissimilarity of clients, $c$ is the number of clients participating at each round, and $k$ is the number of local SGD steps performed by the clients.*

The proof resembles convergence proofs for FedAvg with non-i.i.d. clients, but differs in three aspects that we believe makes our result of independent interest: 1) due to the non-linearity of the hypernetwork, gradients computed from batches might not be unbiased; 2) the updates to the network parameters are not simple averages, but consist of multiple gradient steps on the clients, which the server processes further using the chain rule; 3) the objective includes regularization terms that only the server can compute.

---

[2]Note that all of these operations are standard first-order derivatives and matrix multiplications. In contrast to some meta-learning algorithms, no second-order derivatives, such as gradients of gradients, are required.

**Discussion**   Theorem 3.1 characterizes the convergence rate of PeFLL's optimization step in terms of the number of iterations, $T$, and some problem-specific constants. For illustration, we first discuss the case where the client descriptors are computed from the complete client dataset ($B = S_i$ in Algorithm 1, line 5). In that case, $\sigma_2$ and $\sigma_3$ vanish, such that only three terms remain in (2). The first two are of order $\frac{1}{\sqrt{T}}$, while the third one is of order $\frac{1}{T}$. For sufficiently large $T$, the first two terms dominate, resulting in the same order of convergence as FedAvg (Karimireddy et al., 2020). If clients compute their descriptors from batches ($B \subsetneq S_i$), two additional variance terms emerge in the bound, which depend on the size of the batches used by the clients to compute their descriptors. It is always possible to control these terms, though: for large $S_i$, one can choose $B$ sufficiently large to make the additional terms as small as desired, and for small $S_i$, setting $B = S_i$ is practical, which will make the additional terms disappear completely, see above.

## 3.2   GENERALIZATION GUARANTEES

In this section, we prove a generalization bound that justifies PeFLL's learning objective. Following prior work in learning-to-learn theory we adopt a PAC-Bayesian framework. For this, we assume that the embedding network, the hypernetwork and the personalized models are all learned stochastically. Specifically, the result of training the hypernetwork and embedding network are Gaussian probability distributions over their parameters, $\mathcal{Q}_h = \mathcal{N}(\eta_h; \alpha_h \text{Id})$, and $\mathcal{Q}_v = \mathcal{N}(\eta_v; \alpha_v \text{Id})$, for some fixed $\alpha_h, \alpha_v > 0$. The mean vectors $\eta_h$ and $\eta_v$ are the learnable parameters, and Id is the identity matrix of the respective dimensions. The (stochastic) prediction model for a client with dataset $S$ is obtained by sampling from the Gaussian distribution $Q = \mathcal{N}(\theta; \alpha_\theta \text{Id})$ for $\theta = h(v(S; \eta_v); \eta_h)$ and some $\alpha_\theta > 0$.

We formalize clients as tuples $(D_i, S_i)$, where $D_i$ is their data distribution, and $S_i \overset{i.i.d.}{\sim} D_i$ is training data. For simplicity, we assume that all data sets are of identical size, $m$. We assume that both observed clients and new clients are sampled from a *client environment*, $\mathcal{T}$ (Baxter, 2000), that is, a probability distribution over clients. Then, we can prove the following guarantee of generalization from observed to future clients. For the proof and a more general form of the result, see Appendix B.

**Theorem 3.2.**  *For all $\delta > 0$ the following statement holds with probability at least $1 - \delta$ over the randomness of the clients. For all parameter vectors, $\eta = (\eta_h, \eta_v)$:*

$$\mathop{\mathbb{E}}_{(D,S)\sim\mathcal{T}} \mathop{\mathbb{E}}_{(x,y)\sim D} \mathop{\mathbb{E}}_{\substack{\bar{\eta}_h\sim\mathcal{Q}_h \\ \bar{\eta}_v\sim\mathcal{Q}_v}} \ell\big(x, y, h(v(S; \bar{\eta}_v); \bar{\eta}_h)\big) \leq \frac{1}{n}\sum_{i=1}^{n}\frac{1}{m}\sum_{(x,y)\in S_i} \mathop{\mathbb{E}}_{\substack{\bar{\eta}_h\sim\mathcal{Q}_h \\ \bar{\eta}_v\sim\mathcal{Q}_v}} \ell\big(x, y, h(v(S_i; \bar{\eta}_v); \bar{\eta}_h)\big)$$

$$+ \sqrt{\frac{\frac{1}{2\alpha_h}\|\eta_h\|^2 + \frac{1}{2\alpha_v}\|\eta_v\|^2 + \log(\frac{4\sqrt{n}}{\delta})}{2n}} + \mathop{\mathbb{E}}_{\substack{\bar{\eta}_h\sim\mathcal{Q}_h \\ \bar{\eta}_v\sim\mathcal{Q}_v}}\sqrt{\frac{\frac{1}{2\alpha_\theta}\sum_{i=1}^{n}\|h(v(S_i;\bar{\eta}_v);\bar{\eta}_h)\|^2 + \log(\frac{8mn}{\delta}) + 1}{2mn}} \quad (3)$$

**Discussion**   Theorem 3.2 states that the expected loss of the learned models on future clients (which is the actual quantity of interest) can be controlled by the empirical loss on the observed clients' data (which we can compute) plus two terms that resemble regularizers. The first term penalizes extreme values in the parameters of the hypernetwork and the embedding network. Thereby, it prevents overfitting for the part of the learning process that accumulates information across clients. The second term penalizes extreme values in the output of the hypernetwork, which are the parameters of the per-client models. By this, it prevents overfitting on each client. Because the guarantee holds uniformly over all parameter vectors, we can optimize the right hand side with respect to $\eta$ and the guarantee will still be fulfilled for the minimizer. The PeFLL objective is modeled after the right hand side of (3) to mirror this optimization in simplified form: we drop constant terms and use just the mean vectors of the network parameters instead of sampling them stochastically. Also, we drop the square roots from the regularization terms to make them numerically better behaved.

**Relation to previous work**   A generalization bound of similar form as the one underlying Theorem 3.2 appeared in (Rezazadeh, 2022, Theorem 5.2). That bound, however, is not well suited to the federated setting. First, it contains a term of order $\frac{(n+m)\log m\sqrt{n}}{nm}$, which is not necessarily small for large $n$ (number of clients) but small $m$ (samples per client). In contrast, the corresponding terms in our bound, $\frac{\log\sqrt{n}}{n}$ and $\frac{\log nm}{nm}$, are both small in this regime. Second, when instantiating the bound from Rezazadeh (2022) an additional term of order $\frac{1}{m}\|\eta\|$ would appear, which can be large in the case where the dimensionality of the network parameters is large but $m$ is small.

## 4 EXPERIMENTS

In this section we report on our experimental evaluation[3]. The values reported in every table and plot are given as the mean together with the standard deviation across three random seeds.

**Datasets**     For our experiments, we use three datasets that are standard benchmarks for FL: CI-FAR10/CIFAR100 (Krizhevsky, 2009) and FEMNIST (Caldas et al., 2018). Following prior pFL works, for CIFAR10 and CIFAR100 we form statistically heterogeneous clients by randomly assigning a fixed fraction of the total number of $C$ classes to each of $n$ clients, for $n \in \{100, 500, 1000\}$. The clients then receive test and train samples from only these classes. For CIFAR10 each client has 2 of the $C = 10$ classes and for CIFAR100 each client has 10 of the $C = 100$ classes. FEMNIST is a federated dataset for handwritten character recognition, with $C = 62$ classes (digits and lower/upper case letters) and 817,851 samples. We keep its predefined partition into 3597 clients based on writer identity. We randomly partition the clients into 90% seen and 10% unseen. Seen clients are used for training while unseen clients do not contribute to the initial training and are only used to later assess each method's performance on new clients as described below. Additional experiments on the Shakespeare dataset (Caldas et al., 2018) are provided in Appendix A.

**Baselines**     We evaluate and report results for the following pFL methods, for which we build on the *FL-Bench* repository [4]: Per-FedAvg (Fallah et al., 2020), which optimizes the MAML (Finn et al., 2017) objective in a federated setting; FedRep (Collins et al., 2021), which trains a global feature extractor and per-client classifier heads; pFedMe (Dinh et al., 2020), which trains a personal model per client using a regularization term to penalize differences from a global model; kNN-Per (Marfoq et al., 2022), which trains a single global model which each client uses individually to extract features of their data for use in a $k$-nearest-neighbor-based classifier; pFedHN (Shamsian et al., 2021) which jointly trains a hypernetwork and per client embedding vectors to output a personalized model for each client. For reference, we also include results of (non-personalized) FedAvg (McMahan et al., 2017) and of training a local model separately on each client.

**Constructing models for unseen clients**     We are interested in the performance of PeFLL not just on the training clients but also on clients not seen at training time. As described in Algorithm 1 inference on new clients is simple and efficient for unseen clients as it does not require any model training, by either the client or the server. With the exception of kNN-Per all other methods require some form of finetuning in order to obtain personalized models for new clients. Per-FedAvg and pFedMe obtain personal models by finetuning the global model locally at each client for some small number of gradient steps. FedRep freezes the global feature extractor and optimizes a randomly initialized head locally at each new client. pFedHN freezes the trained hypernetwork and optimizes a new embedding vector for each new client, which requires not just local training but also several communication rounds with the server. The most efficient baseline for inference on a new client is kNN-Per, which requires only a single forward pass through the trained global model and the evaluation of a $k$-nearest-neighbor-based predictor.

**Models**     Following prior works in pFL the personalized model used by each client is a LeNet-style model (Lecun et al., 1998) with two convolutional layers and three fully connected layers. For fair comparison we use this model for PeFLL as well as all reported baselines. PeFLL uses an embedding network and a hypernetwork to generate this personalized client model. For our experiments the hypernetwork is a three layer fully connected network which takes as input a client descriptor vector, $v \in \mathbb{R}^l$, and outputs the parameters of the client model, $\theta \in \mathbb{R}^d$. For further details, see Appendix A. Note that the final layer of the client model predicts across all classes in the dataset. For the embedding network we tested two options, a single linear projection which takes as input a one-hot encoded label vector, and a LeNet-type ConvNet with the same architecture as the client models except that its input is extended by $C$ extra channels that encode the label in one-hot form. The choice of such small models for our embedding network is consistent with the fact that the embedding network must be transmitted to the client. We find that for CIFAR10 and FEMNIST the ConvNet embedding network produces the best results while for CIFAR100 the linear embedding is best, and these are the results we report.

**Training Details**     We train all methods, except Local, for 5000 rounds with partial client participation. For CIFAR10 and CIFAR100 client participation is set to $5\%$ per round. For FEMNIST

---

[3]We provide the code as supplemental material. We will publish it when the anonymity requirement is lifted.
[4]https://github.com/KarhouTam/FL-bench

|  | CIFAR10 | | | CIFAR100 | | | FEMNIST |
|---|---|---|---|---|---|---|---|
| #trn.clients | $n = 90$ | 450 | 900 | 90 | 450 | 900 | 3237 |
| Local | $82.2 \pm 0.6$ | $70.9 \pm 0.5$ | $65.5 \pm 0.7$ | $39.4 \pm 0.2$ | $19.7 \pm 0.2$ | $11.3 \pm 0.1$ | $62.2 \pm 0.1$ |
| FedAvg | $47.5 \pm 0.5$ | $50.4 \pm 0.6$ | $51.9 \pm 0.7$ | $16.4 \pm 0.4$ | $20.2 \pm 0.1$ | $20.2 \pm 0.2$ | $82.1 \pm 0.2$ |
| Per-FedAvg | $79.1 \pm 2.1$ | $76.9 \pm 0.9$ | $76.6 \pm 0.3$ | $40.0 \pm 0.3$ | $31.5 \pm 0.2$ | $20.5 \pm 0.1$ | $82.7 \pm 0.9$ |
| FedRep | $84.6 \pm 0.8$ | $79.2 \pm 0.6$ | $77.3 \pm 0.3$ | $42.7 \pm 0.8$ | $35.5 \pm 0.7$ | $30.8 \pm 0.5$ | $83.6 \pm 0.8$ |
| pFedMe | $83.9 \pm 1.2$ | $79.0 \pm 2.0$ | $80.4 \pm 0.9$ | $39.6 \pm 0.5$ | $40.5 \pm 0.4$ | $34.6 \pm 0.2$ | $85.9 \pm 0.8$ |
| kNN-Per | $84.0 \pm 0.3$ | $82.0 \pm 0.5$ | $79.9 \pm 0.7$ | $42.2 \pm 0.9$ | $38.1 \pm 0.4$ | $34.5 \pm 0.5$ | $85.2 \pm 0.3$ |
| pFedHN | $87.8 \pm 0.5$ | $77.4 \pm 1.3$ | $66.0 \pm 1.1$ | $53.6 \pm 0.2$ | $25.5 \pm 0.3$ | $20.0 \pm 1.2$ | $83.8 \pm 0.3$ |
| PeFLL | $\mathbf{89.0 \pm 0.8}$ | $\mathbf{88.9 \pm 0.4}$ | $\mathbf{87.8 \pm 0.4}$ | $\mathbf{56.0 \pm 0.3}$ | $\mathbf{43.2 \pm 0.8}$ | $\mathbf{40.9 \pm 0.6}$ | $\mathbf{90.1 \pm 0.1}$ |

(a) Accuracy on clients observed at training time (best values per setup in bold)

|  | CIFAR10 | | | CIFAR100 | | | FEMNIST |
|---|---|---|---|---|---|---|---|
| #trn.clients | $n = 90$ | 450 | 900 | 90 | 450 | 900 | 3237 |
| Local | $81.6 \pm 0.7$ | $70.6 \pm 1.8$ | $65.5 \pm 0.5$ | $38.7 \pm 0.9$ | $19.9 \pm 1.1$ | $11.0 \pm 0.5$ | $62.1 \pm 0.2$ |
| FedAvg | $48.6 \pm 0.3$ | $49.6 \pm 1.0$ | $51.7 \pm 0.7$ | $17.1 \pm 0.9$ | $19.3 \pm 2.0$ | $20.5 \pm 1.0$ | $81.9 \pm 0.4$ |
| Per-FedAvg | $77.6 \pm 1.9$ | $69.0 \pm 2.5$ | $67.3 \pm 0.8$ | $35.6 \pm 1.5$ | $30.6 \pm 0.4$ | $20.5 \pm 0.5$ | $81.1 \pm 1.5$ |
| FedRep | $81.8 \pm 1.0$ | $77.4 \pm 1.7$ | $76.0 \pm 0.3$ | $38.6 \pm 1.1$ | $35.9 \pm 1.5$ | $31.8 \pm 1.6$ | $82.8 \pm 0.7$ |
| pFedMe | $78.4 \pm 2.2$ | $74.3 \pm 1.4$ | $72.3 \pm 1.7$ | $31.1 \pm 1.4$ | $32.3 \pm 0.5$ | $28.9 \pm 0.2$ | $86.1 \pm 0.4$ |
| kNN-Per | $82.4 \pm 0.7$ | $80.8 \pm 1.5$ | $78.4 \pm 1.1$ | $\mathbf{41.8 \pm 1.2}$ | $37.2 \pm 2.0$ | $33.7 \pm 0.7$ | $84.6 \pm 0.6$ |
| pFedHN | $63.3 \pm 3.7$ | $60.9 \pm 2.4$ | $57.8 \pm 2.0$ | $24.1 \pm 1.2$ | $21.5 \pm 1.4$ | $20.8 \pm 1.2$ | $82.5 \pm 0.1$ |
| PeFLL | $\mathbf{89.3 \pm 2.2}$ | $\mathbf{88.9 \pm 0.6}$ | $\mathbf{88.5 \pm 0.3}$ | $35.2 \pm 1.4$ | $\mathbf{38.1 \pm 0.4}$ | $\mathbf{38.4 \pm 0.9}$ | $\mathbf{90.7 \pm 0.2}$ |

(b) Accuracy on clients not observed at training time (best values per setup in bold)

Table 1: Experimental results on standard pFL benchmarks. In all settings, except CIFAR100 with 100 clients, PeFLL achieves higher accuracy than previous methods, and the accuracy difference between training clients (top table) and unseen clients (bottom table) is small, if present at all.

we fix the number of clients participating per round to 5. The Local baseline trains on each client independently for 200 epochs. The hyperparameters for all methods are tuned using validation data that was held out from the training set (10,000 samples for CIFAR10 and CIFAR100, spread across the clients, and 10% of each client's data for FEMNIST). The optimizer used for training at the client is SGD with a batch size of 32, a learning rate chosen via grid search and momentum set to 0.9. The batch size used for computing the descriptor is also 32. More details of the hyperparameter selection for each method as well as ablation studies are provided in Appendix A.

## 4.1 RESULTS

Table 1 shows the results for PeFLL and the baseline methods. In all cases, we report the test set accuracy on the clients that were used for training (Table 1a) and on new clients that were not part of the training process (Table 1b). **PeFLL achieves the best results in almost all cases, and often by a large margin.** The improvements over previous methods are most prominent for previously unseen clients, for which PeFLL produces model of almost identical accuracy as for the clients used for training. This is especially remarkable in light of the fact that several of the other methods have computationally more expensive procedures for generating models in this setting than PeFLL, in particular requiring on-client or even federated training to produce the personalized models. We see this result as a strong evidence that PeFLL successful generalizes, as predicted by Theorem 3.2.

Comparing PeFLL's results to the most similar baseline, pFedHN, one observes that the latter's performance decreases noticeably when the number of clients increases and the number of samples per client decrease accordingly. We attribute this to the fact that pFedHN learns independent client descriptors for each client, which can become unreliable if only few training examples are available for each client. Similarly, for Per-FedAvg, pFedMe and FedRep, which construct personalized models by local finetuning, the model accuracy drops when the amount of data per client decreases, especially in the more challenging CIFAR100 setup. kNN-Per maintains good generalization from train to unseen clients, but its performance worsens when the number of samples per client drops and the kNN predictors have less client data available. In contrast, PeFLL's performance remains stable, which we attribute to the fact that it learns the embedding and hypernetwork jointly from all available data. The new client does not have to use its data for training, but rather just to generate a client descriptor. As a pure feed-forward process, this does not suffer from overfitting in the low-data regime.

|  | CIFAR10 | | | CIFAR100 | | | FEMNIST |
|---|---|---|---|---|---|---|---|
| #trn.clients | $n = 90$ | 450 | 900 | 90 | 450 | 900 | 3237 |
| FedAvg | $48.6 \pm 0.3$ | $49.6 \pm 1.0$ | $51.7 \pm 0.7$ | $17.1 \pm 0.9$ | $19.3 \pm 2.0$ | $20.5 \pm 1.0$ | $81.9 \pm 0.4$ |
| PeFLL | $\mathbf{84.5 \pm 0.6}$ | $\mathbf{83.6 \pm 1.0}$ | $\mathbf{81.5 \pm 0.7}$ | $\mathbf{26.3 \pm 0.4}$ | $\mathbf{28.2 \pm 0.6}$ | $\mathbf{30.3 \pm 0.6}$ | $\mathbf{91.4 \pm 0.1}$ |

Table 2: Test accuracy on CIFAR10 and CIFAR100 for (test) clients that have only unlabeled data.

**Personalization for clients with only unlabeled data**     A scenario of particular interest is when some clients have only unlabeled data available. PeFLL can still provide personalized models for them in the following way. To train PeFLL, as before one uses only clients which do have labeled data. However, the embedding network is constructed to take only the unlabeled parts of the client data as input (i.e. it ignores the labels). Consequently, even clients that only have unlabeled data can evaluate the embedding network and obtain descriptors, so the hypernetwork can assign personalized models to them during the test phase. We report results for this in Table 2. Note that none of the personalized baseline methods from Section 4 are able to handle this setting, because they train or finetune on the test clients. Therefore, the only alternative would be to train a non-personalized model, e.g. from FedAvg, and use this for clients with only labeled data. One can see that PeFLL improves the accuracy over this baseline in all cases.

**Analysis of Client Descriptors**     Our work relies on the hypothesis that clients with similar data distributions should obtain similar descriptors, such that the hypernetwork then produces similar models for them. To study this hypothesis empirically, we create clients of different similarities to each other in the following way.

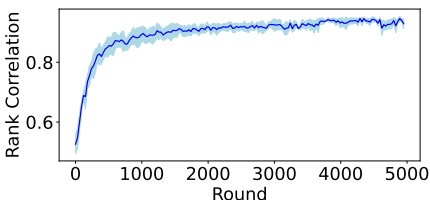

Figure 3: Correlation between *client descriptor similarity* obtained from the embedding network and *ground truth similarity* over the course of training.

Let $C$ denote the number of classes and $n$ the number of clients. Then, for each client $i$ we sample a vector of class proportions, $\pi_i \in \Delta^C$, from a Dirichlet distribution $\text{Dir}(\boldsymbol{\alpha})$ with parameter vector $\boldsymbol{\alpha} = (0.1, \dots, 0.1)$, where $\Delta^C$ is the unitary simplex of dimension $C$. Each client then receives a dataset, $S_i$, of samples from each of the $C$ classes according to its class proportion vector $\pi_i$. We randomly split the clients into train and unseen clients, and we run PeFLL on the train clients. Throughout training we periodically use the embedding network to compute all client descriptors in order to track how they evolve.

For any pair of such clients, we compute two similarity measures: $d_{ij}^\pi = \|\pi_i - \pi_j\|$, which provides a ground truth notion of similarity between client distributions, and $d_{ij}^v = \|v(S_i) - v(S_j)\|$, which measures the similarity between client descriptors. To measure the extent to which $d^v$ reflects $d^\pi$, we use their average rank correlation in the following way. For any unseen client $i$ we a form vector of similarities $d_i^v = (d_{i1}^v, \dots, d_{in}^v) \in \mathbb{R}^n$, and compute its Spearman's rank correlation coefficient (Spearman, 1904) to the corresponding ground truth vector $d_i^\pi = (d_{i1}^\pi, \dots, d_{in}^\pi) \in \mathbb{R}^n$. Figure 3 shows the average of this value across the unseen clients after different numbers of training steps (CIFAR 10 dataset, 1000 clients in total). One can see that the rank correlation increases over the course of training, reaching very high correlation values of approximately 0.93. Results for other numbers of clients are comparable, see Appendix A. This indicates that the embedding network indeed learns to organize the descriptor space according to client distributional similarity, with similar clients being mapped to similar descriptors. Moreover, because these results are obtained from unseen clients, it is clear that the effect does not reflect potential overfitting of the embedding network to the training clients, but that the learned similarity indeed generalizes well.

## 5  CONCLUSION

In this work, we presented PeFLL, a new algorithm for personalized federated learning based on learning-to-learn. By means of an embedding network that creates inputs to a hypernetwork it efficiently generates personalized models, both for clients present during training and new clients that appear later. PeFLL has several desirable properties for real-world usability. It stands on solid theoretical foundation in terms of convergence and generalization. It is trained by stateless optimization and does not require additional training steps to obtain models for new clients. Overall, it has low latency and puts little computational burden on the clients, making it practical to use in large scale applications with many clients each possessing little data.

**Acknowledgements.** This research was supported by the Scientific Service Units (SSU) of ISTA through resources provided by Scientific Computing (SciComp).

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

| Layer | Shape | Nonlinearity |
|---|---|---|
| Conv1 | $3 \times 5 \times 5 \times 16$ | ReLU |
| MaxPool | $2 \times 2$ | – |
| Conv2 | $16 \times 5 \times 5 \times 32$ | ReLU |
| MaxPool | $2 \times 2$ | flatten |
| FC1 | $800 \times 120$ | ReLU |
| FC2 | $120 \times 84$ | ReLU |
| FC3 | $84 \times C$ | softmax |

| Layer | Shape | Nonlinearity |
|---|---|---|
| Conv1 | $3 \times 5 \times 5 \times 16$ | ReLU |
| MaxPool | $2 \times 2$ | – |
| Conv2 | $16 \times 5 \times 5 \times 32$ | ReLU |
| MaxPool | $2 \times 2$ | flatten |
| FC1 | $800 \times 120$ | ReLU |
| FC2 | $120 \times 84$ | ReLU |
| FC3 | $84 \times l$ | none |

Table 3: LeNet-based ConvNet of personalized client models. Convolutions are without padding. $C$ is the number of classes.

Table 4: LeNet-based embedding network. Convolutions are without padding. $l$ is the dimensionality of the client descriptor.

| Layer | Shape | Nonlinearity |
|---|---|---|
| FC1 | $l \times 100$ | ReLU |
| FC2 | $100 \times 100$ | ReLU |
| FC3 | $100 \times 100$ | ReLU |
| FC4 | $100 \times 100$ | ReLU |
| FC5 | $100 \times D$ | none |

Table 5: Fully connected hypernetwork. $D$ is the dimensionality of the model to be created. Note that because of FC5 alone the hypernetwork is at least 100 bigger than the client models.

# A  APPENDIX - EXPERIMENTS

## A.1  MODEL ARCHITECTURES

We used a LeNet-based architecture for the personalized model on each client. The exact architecture is shown in Table 3. After each convolutional and fully connected layer we apply a ReLU non-linearity except for the final layer which uses a softmax for input into the cross-entropy loss. The embedding network uses the same architecture except that the final layer has dimension $84 \times l$ and we do not apply a non-linearity (Table 4). For the hypernetwork we use a fully connected network. The exact architecture is shown in Table 5. Again we apply a ReLU after each layer except the last one, which does not have a non-linearity and simply has output of size $D$, where $D$ denotes the number of parameters in the client personalized model.

## A.2  HYPERPARAMETER SETTINGS

For PeFLL, each client uses a single batch of data (batch size 32) as input to the embedding network. For the hypernetwork parameter settings we use the recomended values given in (Shamsian et al., 2021) as we found these to work well in our setting as well. Specifically, the dimension of the embedding vectors is $l = \frac{n}{4}$ and the number of client SGD steps is $k = 50$. The regularization parameters for the embedding network and hypernetwork are set to $\lambda_h = \lambda_v = 10^{-3}$, while the output regularization is $\lambda_\theta = 0$.

For Per-FedAvg we used the first order (FO) approximation detailed in (Fallah et al., 2020), and following the recomendations we set the number of client local steps is set to be a single epoch. For FedRep we set the number of client updates to the head and body to 5 and 1 epochs respectively as recommended in (Collins et al., 2021). To infer on a new client we we randomly initialized a head and trained only the head locally for 20 epochs. For pFedMe we follow the recomendations in (Dinh et al., 2020) and set the regularization parameter to $\lambda = 15$, the computation parameter to $K = 3$ and $\beta = 1$. To infer for a new client we first initialize the clients personalized model to the global model and finetuned this for 20 epochs using the pFedMe loss function. For kNN-Per, for the scale parameter in the kNN classifier we searched over the values recommended in (Marfoq et al., 2022), namely $\{0.1, 1, 10, 100\}$, we found that setting the scale parameter to 100 to worked best. The $k$ parameter for the kNN search and the weight parameter for how much to weight the client local vs global model were chosen individually on each client by cross-validation searching over $\{5, 10\}$ and $\{0.0, 0.1, 0.2, \ldots, 1.0\}$ respectively. For pFedHN we set the hyperparameters following the recommendations in (Shamsian et al., 2021). For inferring on a new client we followed the

procedure given in (Shamsian et al., 2021) and we randomly initialized a new embedding vector and optimized only this using 20 rounds of client server communication.

### A.3 ADDITIONAL EXPERIMENTS

**Comparison of communication and computation cost with FedAvg** We compare the performance of PeFLL to FedAvg with respect to communication costs of training. At each global round, the communication cost of PeFLL between the server and $c$ clients is $2c(\|\theta\| + \|\eta_v\| + \|v\|)$ while for FedAvg is $2c\|\theta\|$. In our setting, the embedding network and client models are of comparable size and the size of client descriptors ($v$) is negligibly small compared to them. Therefore, the per round communication cost of PeFLL is about twice that of FedAvg. In Figures 8–14 we plot the accuracies during training for a given communication cost (measured in GB of information transferred between server and clients) as well computational cost for the clients (measured in number of global rounds). As shown in the figures, for any given communication or computational budget PeFLL outperforms FedAvg on these datasets.

**Shakespeare dataset** In order to test the effectiveness of PeFLL in non-image tasks, we also provide results for the Shakespeare dataset (Caldas et al., 2018), which is a next character prediction task. We use the LSTM model from Caldas et al. (2018) as the client model, and as the embedding network. The hypernetwork is the same as in all other experiments. We compare the performance in comparison to a number of baselines. Figure 5 shows the comparison of the accuracies for train and test clients during training with respect to the number of global rounds. PeFLL outperforms all baselines when comparing performance with respect to the number of global rounds. We additionally compare the performance of PeFLL with FedAvg with respect to accuracy for a given communication cost, see Figure 6. We observe similar performances for a fixed communication budget.

**ResNet Experiments** In order to test the effectiveness of PeFLL for more modern architectures, we also performed experiments using a ResNet20 model[5] for CIFAR10 dataset. We compared the performance of PeFLL with FedAvg, and the results are shown in Figure 7 for 100, 500, and 1000 clients, respectively. These figures show PeFLL outperforms FedAvg also for this model, and support the feasibility of generating modern models as part of our pipeline.

**Analysis of Client Extrapolation** For all experiments so far, clients at training time and new clients were related in the sense that they come from the same client environment. In this section, we examine how well PeFLL is able to generalize beyond this, by studying its *extrapolation* performance, where the new clients come from a different client environment than the train clients. We follow the same procedure as in the previous section to simulate heterogeneous clients and use sampled Dirichlet class proportions to assign classes. For the train clients we again use $\boldsymbol{\alpha}_{\text{train}} = (0.1, \ldots, 0.1)$ to generate each clients class proportion vector. However, for the new clients we use a different Dirichlet parameter $\boldsymbol{\alpha}_{\text{new}} = (\alpha, \ldots, \alpha)$. For each $\alpha \in \{0.1, 0.2, 0.3, \ldots, 1.0\}$ we generate a group of new clients using this parameter. We run PeFLL on the train clients then use the trained embedding and hypernetworks to generate a model for each of the new clients.

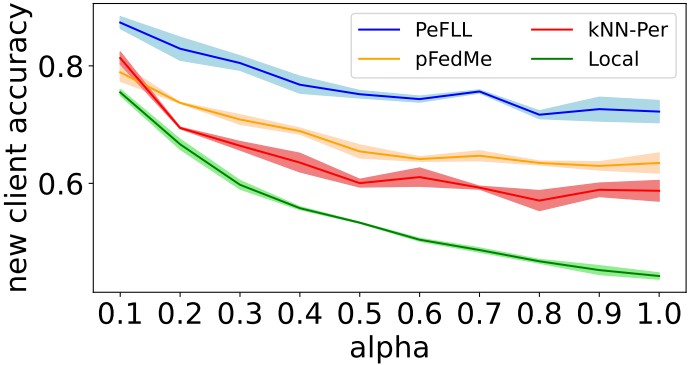

Figure 4: Accuracy in *client extrapolation*. Larger values of $\alpha$ indicate new clients that are more dissimilar to the train clients.

[5]We use the implementation from `https://github.com/chenyaofo/pytorch-cifar-models`, except without the batch normalization layers.

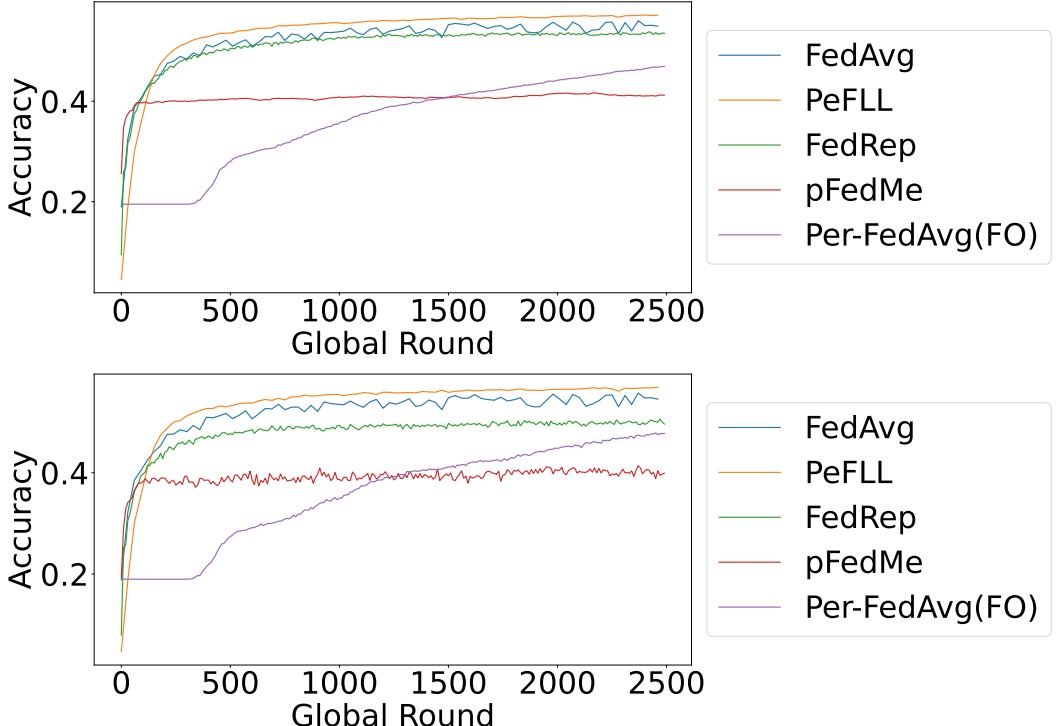

Figure 5: Test Accuracies for *train* clients (top row) and *test* clients (bottom row) during different steps of the training for PeFLL and baselines, for the Shakespeare dataset.

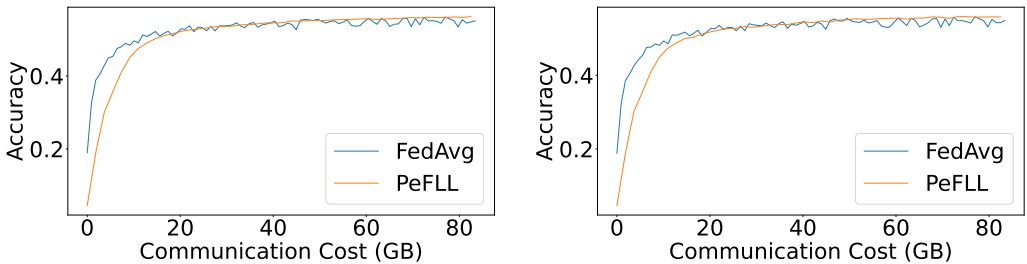

Figure 6: Test Accuracies for *train* clients (left) and *test* clients (right) over the course of training measured with respect to total communication cost for PeFLL and FedAvg, for the Shakespeare dataset.

Figure 4 shows the resulting accuracy values for PeFLL and the best performing baselines of Table 1, as well the baseline of purely local per-client training. Note that as $\alpha$ increases so does the difficulty of the client's problem, as illustrated by the fact that the accuracy of purely local training decreases. Despite this increased task difficulty and distributional difference PeFLL still obtains strong results. Even at $\alpha = 1$, PeFLL produces models that are more accurate than those learned by the other methods for smaller values of $\alpha$, and far superior to purely local training on the new clients.

**Integrating known client label sets**    In the vanilla setting of Section 4, only the client training data is used to create a model. By design, each model was a multi-class classifier across all possible classes, even if only a subset of those are relevant for the client. However, it is also reasonable to assume that clients might have prior knowledge which of the output classes they actually want to predict. This information can be included by *label masking (LM)*, in which the client overwrites the output probabilities of classes that cannot occur for it by 0, such that they will never be predicted. We report results for this in Table 6. One can see that this has minor effects for CIFAR10, but the results for CIFAR100 are clear improved. We attribute this to the fact that the larger label set and the smaller number of examples per class increase the chances of spurious classes being predicted.

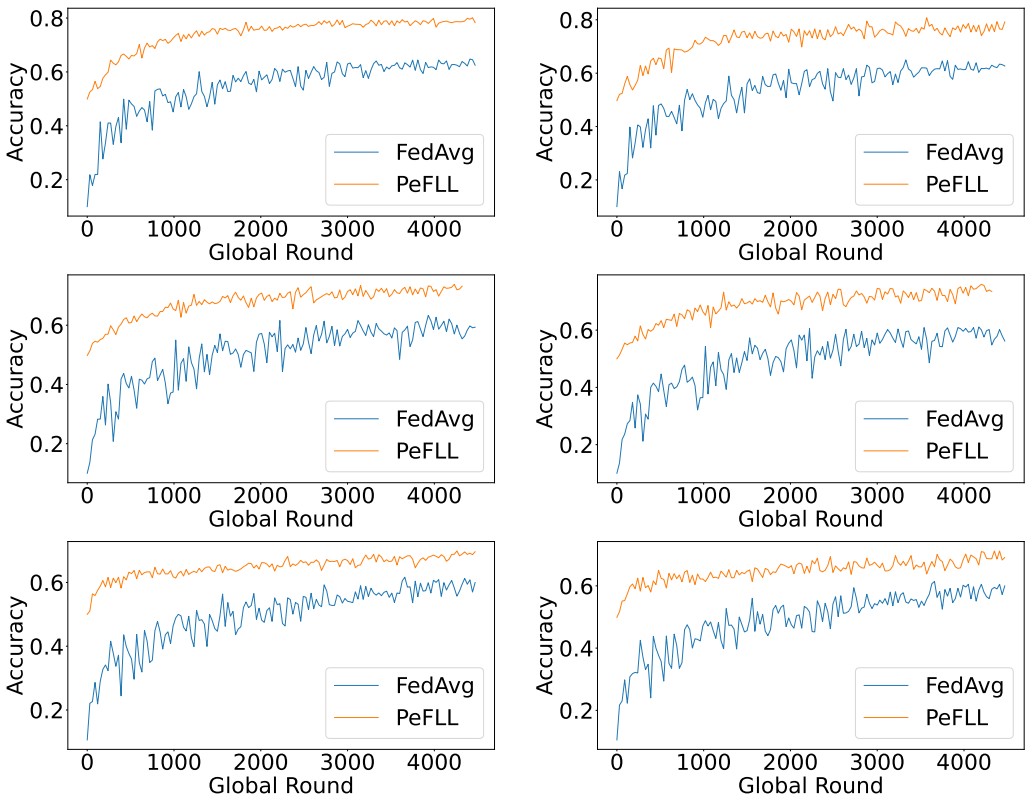

Figure 7: Test Accuracies for *train* clients (left) and *test* clients (right) during different steps of the training for PeFLL and FedAvg for the CIFAR10 dataset with 100 clients (top row), 500 clients (middle row) and 1000 clients (bottom row). PeFLL consistently outperforms FedAvg.

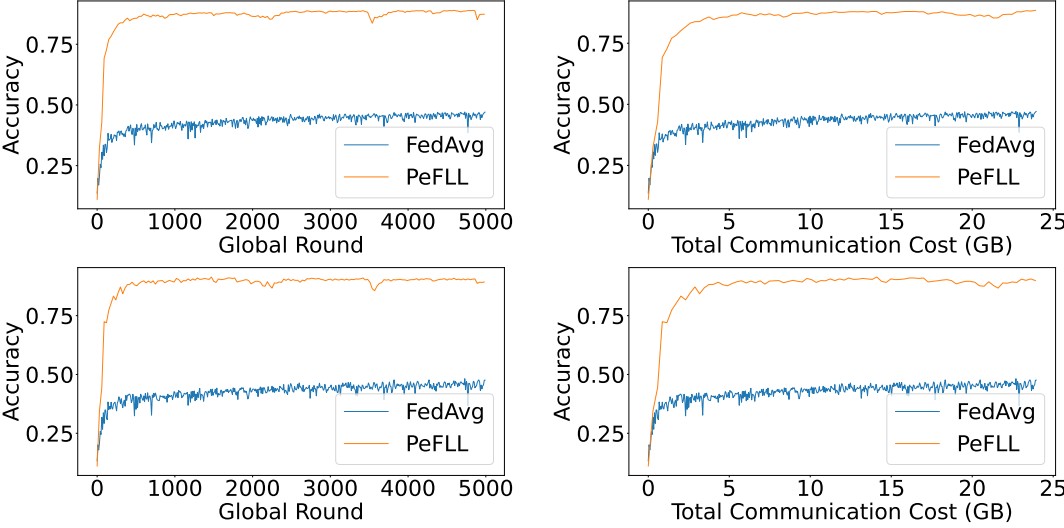

Figure 8: Test Accuracies for *train* clients (top) and *test* clients (bottom) during different steps of the training for PeFLL and FedAvg, for the CIFAR10 dataset, 100 clients. PeFLL's accuracy is higher than FedAvg's with respect to the number of rounds (which roughly reflect the computational cost for the clients) as well as the communication cost.

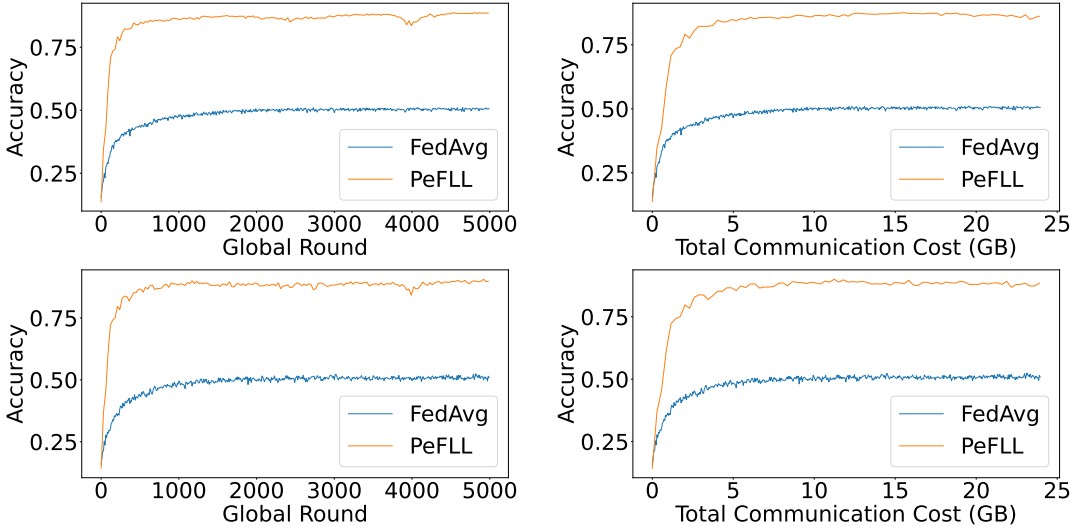

Figure 9: Test Accuracies for *train* clients (top) and *test* clients (bottom) during different steps of the training for PeFLL and FedAvg, for the CIFAR10 dataset, 500 clients. PeFLL's accuracy is higher than FedAvg's with respect to the number of rounds (which roughly reflect the computational cost for the clients) as well as the communication cost.

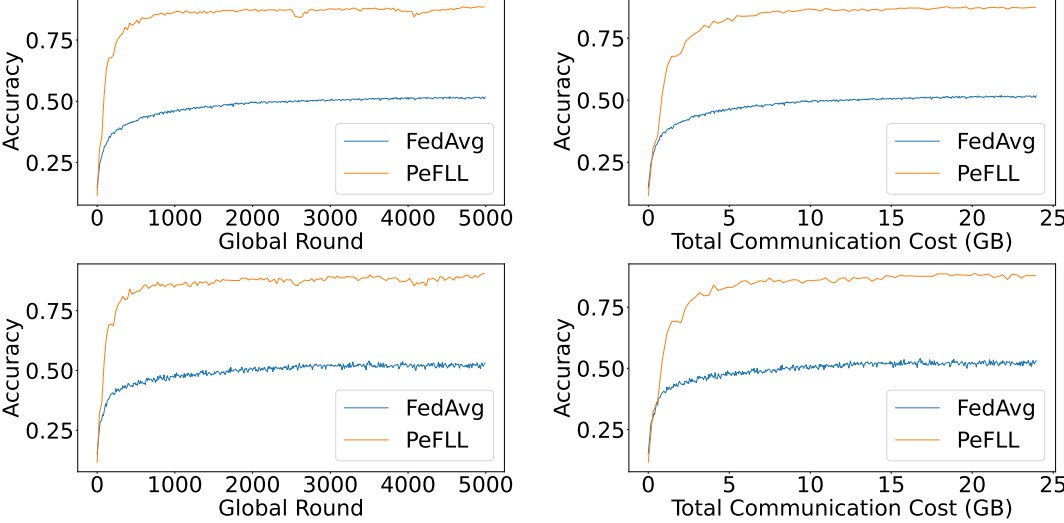

Figure 10: Test Accuracies for *train* clients (top) and *test* clients (bottom) during different steps of the training for PeFLL and FedAvg, for the CIFAR10 dataset, 1000 clients. PeFLL's accuracy is higher than FedAvg's with respect to the number of rounds (which roughly reflect the computational cost for the clients) as well as the communication cost.

## A.4 ADDITIONAL FIGURES

We include additional figures for the correlation between the distances between client descriptors and the true client distributional distances as described in Section 4. In Figures 15 - 17 we show the results of the experiment for different numbers of clients, 100, 500 and 1000. As we can see in all cases high correlation is achieved. With more clients we obtain higher levels of correlation and less variability in the results.

## A.5 ABLATION STUDIES

We include several ablation studies to explore how different design choices impact PeFLL's performance.

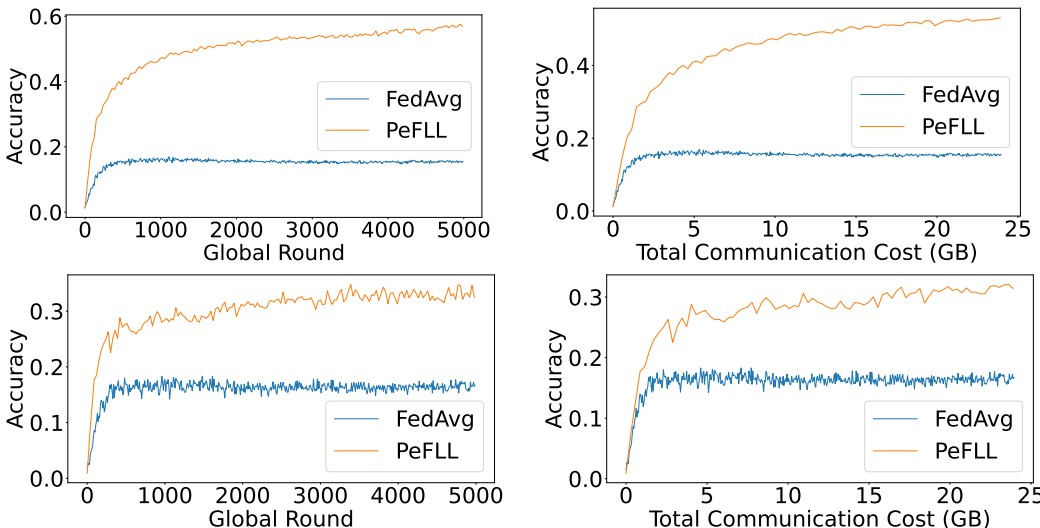

Figure 11: Test Accuracies for *train* clients (top) and *test* clients (bottom) during different steps of the training for PeFLL and FedAvg, for the CIFAR100 dataset, 100 clients. PeFLL's accuracy is higher than FedAvg's with respect to the number of rounds (which roughly reflect the computational cost for the clients) as well as the communication cost.

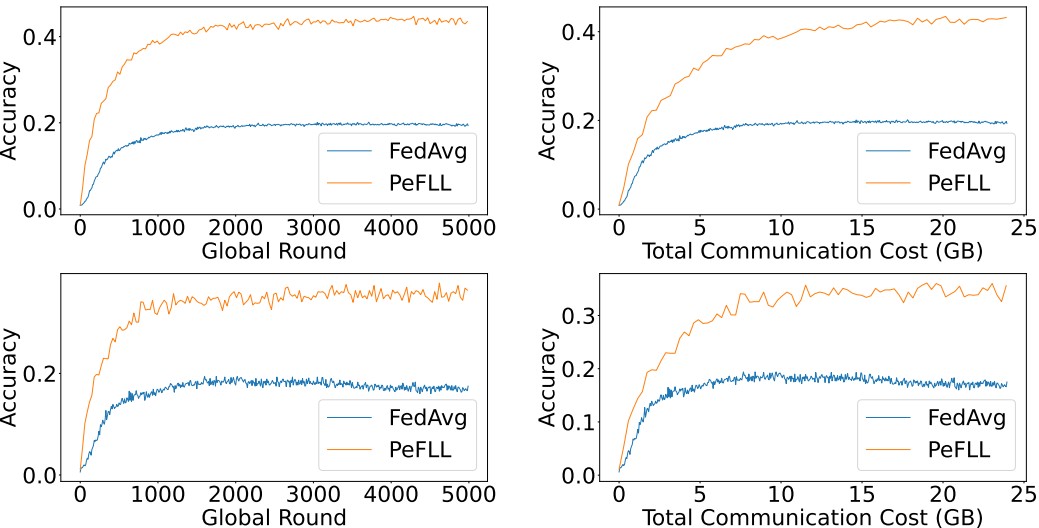

Figure 12: Test Accuracies for *train* clients (top) and *test* clients (bottom) during different steps of the training for PeFLL and FedAvg, for the CIFAR100 dataset, 500 clients. PeFLL's accuracy is higher than FedAvg's with respect to the number of rounds (which roughly reflect the computational cost for the clients) as well as the communication cost.

**Hypernetwork Size.** We investigate how the size of the hypernetwork impacts performance. We keep the same hypernetwork architecture as described in Table 5 but vary the number of hidden layers. We have a small network (S) with only a single hidden layer, a medium network (M) with 3 hidden layers and a large network (L) with 5 hidden layers. We evaluate on CIFAR10 and CIFAR100 using the same experimental setup as for previous experiments, with 100, 500 and 1000 clients. The results are shown in Table 7. As we can see in the results, for CIFAR10 all 3 networks are reasonably similar with a slight drop in performance observed for the S network. For CIFAR100 it is more pronounced, with a modest increase in performance for the L network and a noticeable drop for the S network.

**Regularization Parameters.** We examine the impact of varying the regularization parameters of the objective (1) on the performance of PeFLL. We consider different values for the regularization parameters and study the values in $\{0, 5 \times 10^{-5}, 5 \times 10^{-3}, 5 \times 10^{-1}\}$ for client regularization parameter,

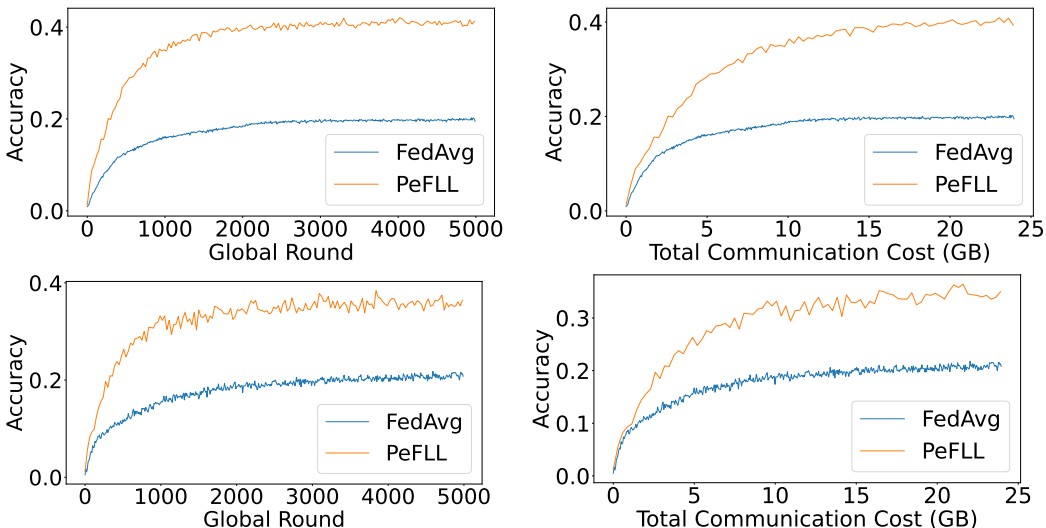

Figure 13: Test Accuracies for *train* clients (top) and *test* clients (bottom) during different steps of the training for PeFLL and FedAvg, for the CIFAR100 dataset, 1000 clients. PeFLL's accuracy is higher than FedAvg's with respect to the number of rounds (which roughly reflect the computational cost for the clients) as well as the communication cost.

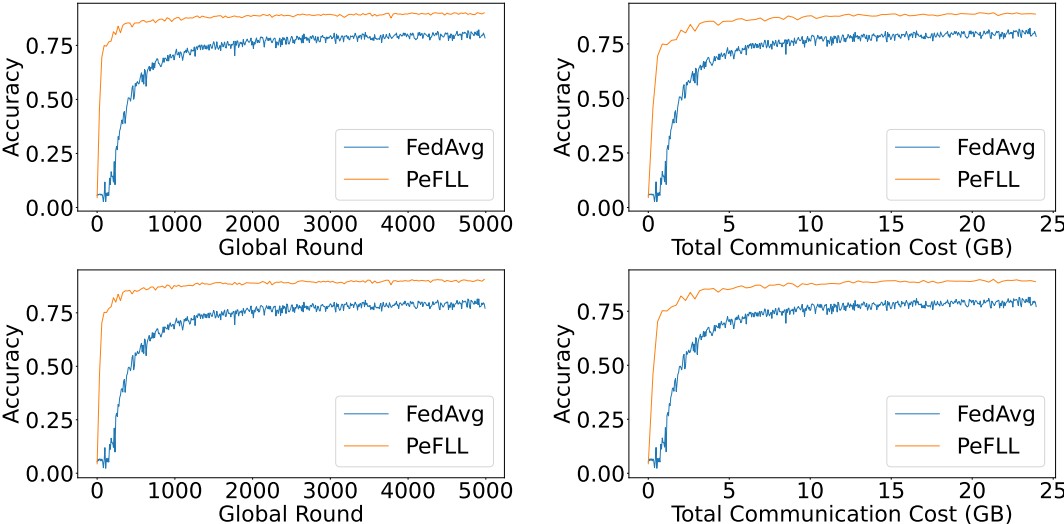

Figure 14: Test Accuracies for *train* clients (top) and *test* clients (bottom) during different steps of the training for PeFLL and FedAvg, for the FEMNIST dataset. PeFLL's accuracy is higher than FedAvg's with respect to the number of rounds (which roughly reflect the computational cost for the clients) as well as the communication cost.

and values in $\{0, 5 \times 10^{-5}, 5 \times 10^{-3}\}$ for embedding and hypernetwork regularization. The results are shown in Table 8. As one can see, PeFLL's performance is rather robust over the regularization strengthh. However, consistent with the theory, modest amounts of hypernetwork regularization parameter help preventing overfitting on training clients and improve client-level generalization. Similarly, modest amounts of client regularization improve sample-level generalization. As can be expected, very large values of this parameter leads to underfitting.

**Embedding network architecture.**    We also study the effect of different architecture choices for the embedding network. We compare the performance of a single-layer MLP network layer and a LeNet-style ConvNet for CIFAR10 and CIFAR100. As we can see in Table 9 the CNN embedding outperforms the MLP in all experiments.

|  | CIFAR10 | | | CIFAR100 | | |
|---|---|---|---|---|---|---|
| #trn.clients | $n = 90$ | 450 | 900 | 90 | 450 | 900 |
| PeFLL | $89.0 \pm 0.8$ | $88.9 \pm 0.4$ | $87.8 \pm 0.4$ | $56.0 \pm 0.3$ | $43.2 \pm 0.8$ | $40.9 \pm 0.6$ |
| PeFLL + LM | $88.9 \pm 0.7$ | $88.9 \pm 0.3$ | $88.4 \pm 0.4$ | $59.5 \pm 0.7$ | $53.2 \pm 0.6$ | $51.4 \pm 0.3$ |
|  | CIFAR10 | | | CIFAR100 | | |
| #trn.clients | $n = 90$ | 450 | 900 | 90 | 450 | 900 |
| PeFLL | $89.3 \pm 2.2$ | $88.9 \pm 0.6$ | $88.5 \pm 0.3$ | $35.2 \pm 1.4$ | $38.1 \pm 0.4$ | $38.4 \pm 0.9$ |
| PeFLL + LM | $88.9 \pm 2.4$ | $88.2 \pm 0.9$ | $87.4 \pm 1.3$ | $55.6 \pm 1.6$ | $53.4 \pm 0.1$ | $51.1 \pm 0.2$ |

Table 6: Test accuracy on CIFAR10 and CIFAR100 for training clients (top) and test clients (bottom) without and with label masking (LM).

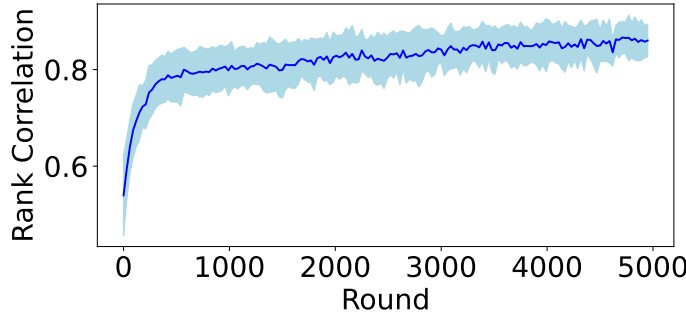

Figure 15: Correlation between *client descriptor similarity* obtained from the embedding network and *ground truth similarity* over the course of training. CIFAR10, 100 Clients.

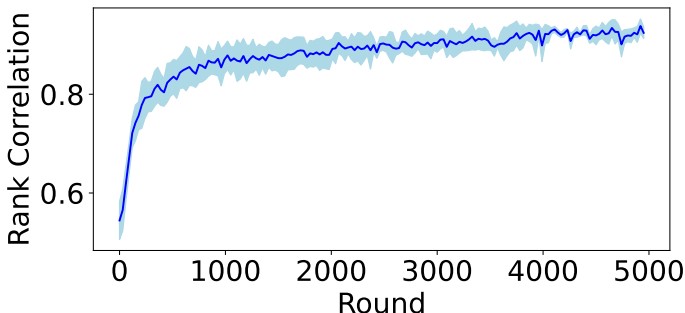

Figure 16: Correlation between *client descriptor similarity* obtained from the embedding network and *ground truth similarity* over the course of training. CIFAR10, 500 clients.

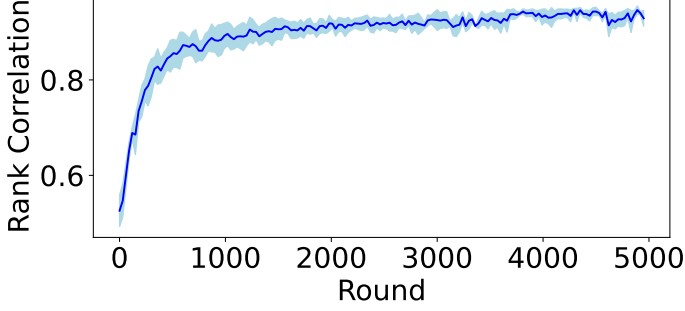

Figure 17: Correlation between *client descriptor similarity* obtained from the embedding network and *ground truth similarity* over the course of training. CIFAR10, 1000 clients.

|  | CIFAR10 | | | CIFAR100 | | |
|---|---|---|---|---|---|---|
| #Clients | 100 | 500 | 1000 | 100 | 500 | 1000 |
| S | $88.7 \pm 0.3$ | $88.2 \pm 0.4$ | $87.7 \pm 0.4$ | $57.9 \pm 0.3$ | $51.9 \pm 0.6$ | $47.4 \pm 1.4$ |
| M | $88.9 \pm 0.7$ | $88.9 \pm 0.3$ | $88.1 \pm 0.3$ | $59.4 \pm 0.2$ | $53.1 \pm 0.4$ | $50.8 \pm 0.5$ |
| L | $88.7 \pm 0.8$ | $88.8 \pm 0.3$ | $88.2 \pm 0.6$ | $59.8 \pm 0.1$ | $53.8 \pm 0.2$ | $52.6 \pm 1.0$ |

Table 7: Test accuracy on CIFAR10 and CIFAR100 using different sizes of hypernetworks, small (S), medium (M) and large (L).

|  | $\lambda_h = \lambda_v$ | | |
|---|---|---|---|
| $\lambda_\theta$ | 0 | $10^{-5}$ | $10^{-3}$ |
| 0 | $87.4 \pm 1.1$ | $87.4 \pm 1.0$ | $88.4 \pm 0.6$ |
| $5 \cdot 10^{-5}$ | $88.7 \pm 0.7$ | $88.7 \pm 0.6$ | $88.8 \pm 0.3$ |
| $5 \cdot 10^{-3}$ | $87.8 \pm 0.8$ | $87.6 \pm 0.8$ | $87.5 \pm 0.5$ |
| $5 \cdot 10^{-1}$ | $82.5 \pm 1.5$ | $83.8 \pm 1.1$ | $86.2 \pm 0.4$ |

Table 8: Test accuracy on CIFAR10 with 100 clients for different settings of the regularization parameters. On the left is the regularization of the client network, and along the top is the regularization of the hypernetwork and embedding network.

|  | CIFAR10 | | | CIFAR100 | | |
|---|---|---|---|---|---|---|
| #Clients | 100 | 500 | 1000 | 100 | 500 | 1000 |
| MLP | $88.5 \pm 1.2$ | $87.8 \pm 1.1$ | $87.5 \pm 2.1$ | $59.1 \pm 0.2$ | $53.4 \pm 0.4$ | $51.2 \pm 0.5$ |
| CNN | $88.9 \pm 0.7$ | $88.9 \pm 0.3$ | $88.4 \pm 0.4$ | $59.4 \pm 0.2$ | $54.4 \pm 0.3$ | $53.2 \pm 0.3$ |

Table 9: Test accuracy on CIFAR10 and CIFAR100 using different embedding network architectures. MLP is a single hidden layer fully connected network and CNN is a LeNet-style ConvNet.

# B    APPENDIX - GENERALIZATION

In this section we prove a new generalization bound for the generic learning-to-learn setting that, in particular, will imply Theorem 3.2 as a special case. Before formulating and proving our main results, we remind the reader of the PAC-Bayesian learning framework (McAllester, 1998) and its use for obtaining guarantees for learning-to-learn.

**PAC-Bayesian learning and learning-to-learn**    In standard PAC-Bayesian learning, we are given a set of possible models, $\mathcal{H}$, and a prior distribution over these $P \in \mathcal{M}(\mathcal{H})$, where $\mathcal{M}(\cdot)$ denotes the set of probability measures over a base set. Given a dataset, $S$, *learning a model* means constructing a new (posterior) distribution, $Q \in \mathcal{M}(\mathcal{H})$, which is meant to give high probability to models with small loss. The posterior distribution, $Q$, induces a stochastic predictor: for any input, one samples a specific model, $f \sim Q$, and outputs the result of this model applied to the input. Note that $Q$ can in principle be a Dirac delta distribution at a single model, resulting in a deterministic predictor. However, for large (uncountably infinite) model such a choice typically does not lead to strong generalization guarantees. For conciseness of notation, in the following we do not distinguish between the distributions over models and their stochastic predictors.

The quality of a stochastic predictor, $Q$, on a data point $(x, y)$ is quantified by its expected loss, $\ell_{(x,y)}(Q) = \mathbb{E}_{f \sim Q} \ell(x, y, f)$. From this, we define the empirical error on a dataset, $S$, as $\frac{1}{|S|} \sum_{(x,y) \in S} \ell_{(x,y)}(Q)$, and its expected loss with respect to a data distribution, $D$, as $\mathbb{E}_{(x,y) \sim D} \ell_{(x,y)}(Q)$. Ordinary PAC-Bayesian generalization bounds provide high-probability upper bounds to the expected loss of a stochastic predictors by the corresponding empirical loss as well as some complexity terms, which typically include the Kullback-Leibler divergence between the chosen posterior distribution and the original (training-data independent) prior, $D_{\mathrm{KL}}(Q||P)$ (McAllester, 1998).

Typically, the posterior distribution is not chosen arbitrarily, but it is the result of a *learning algorithm*, $A : (\mathcal{X} \times \mathcal{Y})^m \to \mathcal{M}(\mathcal{H})$, which takes as input the training data and, potentially, the prior distribution. The idea of *learning-to-learn* (also called *meta-learning* or *lifelong learning* in the literature) is to *learn the learning algorithm* (Baxter, 2000; Pentina and Lampert, 2015).

To study this theoretically, we adopt the setting where $n$ learning tasks is available, which we write as tuples, $(S_i, D_i)$, for $i \in \{1, \ldots, n\}$, each with a data set $S_i \subset \mathcal{X} \times \mathcal{Y}$ that is sampled from a corresponding data distribution $D_i \in \mathcal{M}(\mathcal{X} \times \mathcal{Y})$. For simplicity, we assume that all datasets are of the same size, $m$. We assume tasks are sampled i.i.d. from a *task environment*, $\mathcal{T}$, which is simply a data distribution over such tuples.

Again adopting the PAC-Bayesian framework, we assume that a data-independent (meta-)prior distribution over learning algorithms is available, $\mathcal{P} \in \mathcal{M}(\mathcal{A})$, where $\mathcal{A}$ is the set of possible algorithms, and the goal is use the observed task data to construct a (meta-)posterior distribution, $\mathcal{Q} \in \mathcal{M}(\mathcal{A})$. As before, the resulting procedure is stochastic: at every invocation, the system samples an algorithm $A \sim \mathcal{Q}$. It applies this to the training data, obtaining a posterior distribution $A(S)$, and it makes predictions by sampling models accordingly, $f \sim A(S)$.

Analogously to above situation, we define two measures of quality for such a stochastic algorithms. Its *empirical loss on the data of the observed clients*:

$$\widehat{\mathrm{er}}(\mathcal{Q}) := \frac{1}{n} \sum_{i=1}^{n} \mathbb{E}_{A \sim \mathcal{Q}} \frac{1}{m} \sum_{j=1}^{m} \ell(x_j^i, y_j^i, A(S_i)), \tag{4}$$

and its *expected loss on future clients,*

$$\mathrm{er}(\mathcal{Q}) := \mathbb{E}_{(D,S) \sim \mathcal{T}} \mathbb{E}_{A \sim \mathcal{Q}} \mathbb{E}_{(x,y) \sim D} \ell(x, y, A(S)). \tag{5}$$

The following theorem provides a connection between both quantities.

**Theorem B.1.** *Let $\mathcal{P} \in \mathcal{M}(\mathcal{A})$ and $P \in \mathcal{M}(\mathcal{H})$ be a meta-prior and a prior distribution, respectively, which are chosen independently of the observed training data, $S_1, \ldots, S_n$. Assume that the loss function is bounded in $[0, M]$. Then, for all $\delta \geq 0$ it holds with probability at least $1 - \delta$ over the*

*sampling of the datasets, that for all distributions $\mathcal{Q} \in \mathcal{M}(\mathcal{A})$ over algorithms,*

$$er(\mathcal{Q}) \leq \widehat{er}(\mathcal{Q}) + M\sqrt{\frac{D_{\mathrm{KL}}(\mathcal{Q}||\mathcal{P}) + \log(\frac{4\sqrt{n}}{\delta})}{2n}} + M \underset{A \sim \mathcal{Q}}{\mathbb{E}} \sqrt{\frac{\sum_{i=1}^{n} D_{\mathrm{KL}}(A(S_i)||P) + \log(\frac{8mn}{\delta}) + 1}{2mn}}$$

(6)

where $D_{\mathrm{KL}}$ denotes the Kullback-Leibler divergence.

*Proof.* The proof resembles previous proofs for PAC-Bayesian learning-to-learn learning, but it differs in some relevant steps. We discuss this in a *Discussion* section after the proof itself.

First, we observe that (6) depends linearly in $M$. Therefore, if suffices to prove the case of $M = 1$, and the general case follows by applying the theorem to $\ell/M$ with subsequent rescaling. Next, we define the *expected loss for the $n$ training clients*, $\widetilde{er}(\mathcal{Q})$, as an intermediate object:

$$\widetilde{er}(\mathcal{Q}) = \frac{1}{n}\sum_{i=1}^{n} \underset{A \sim \mathcal{Q}}{\mathbb{E}} \underset{(x,y) \sim D_i}{\mathbb{E}} \ell(x, y, A(S_i)). \tag{7}$$

To bound $er(\mathcal{Q}) - \widehat{er}(\mathcal{Q})$, we divide it in two parts: $er(\mathcal{Q}) - \widetilde{er}(\mathcal{Q})$ and $\widetilde{er}(\mathcal{Q}) - \widehat{er}(\mathcal{Q})$. We then bound each part separately and combine the results.

**Part I** For the former, for any $h \in \mathcal{H}$ let $\Delta_i(h) = \frac{1}{m}\sum_{j=1}^{m} \ell(x_j^i, y_j^i, h) - \mathbb{E}_{(x,y) \sim D_i} \ell(x, y, h)$, and $\Delta_i(Q) = \mathbb{E}_{h \in Q} \Delta_i(h)$, such that $\widetilde{er}(\mathcal{Q}) - \widehat{er}(\mathcal{Q}) = \mathbb{E}_{A \sim \mathcal{Q}} \frac{1}{n}\sum_{i=1}^{n} \Delta_i(A(S_i))$. Then, for any $\lambda > 0$,

$$\widetilde{er}(\mathcal{Q}) - \widehat{er}(\mathcal{Q}) - \frac{1}{\lambda} \underset{A \sim \mathcal{Q}}{\mathbb{E}} D_{\mathrm{KL}}(A(S_1) \times \cdots \times A(S_n) || P \times \cdots \times P) \tag{8}$$

$$\leq \sup_{Q_1,\dots,Q_n} \frac{1}{n}\sum_{i=1}^{n} \Delta_i(Q_i) - \frac{1}{\lambda} D_{\mathrm{KL}}(Q_1 \times \cdots \times Q_n || P \times \dots \times P) \tag{9}$$

$$\leq \frac{1}{\lambda} \log \underset{h_1 \sim P}{\mathbb{E}} \cdots \underset{h_n \sim P}{\mathbb{E}} \prod_{i=1}^{n} e^{\frac{\lambda}{n}\Delta_i(h_i)}, \tag{10}$$

where the second inequality is due to the *change of measure inequality* (Seldin et al., 2012).

Because for each $i = 1, \dots, n$, $P$ is independent of $S_i, \dots, S_n$, we have

$$\underset{S_1,\dots,S_n}{\mathbb{E}} \underset{h_1 \sim P}{\mathbb{E}} \cdots \underset{h_n \sim P}{\mathbb{E}} \prod_{i=1}^{n} e^{\frac{\lambda}{n}\Delta_i(h_i)} = \underset{S_1}{\mathbb{E}} \underset{h_1 \sim P}{\mathbb{E}} e^{\frac{\lambda}{n}\Delta_1(h_1)} \cdots \underset{S_n}{\mathbb{E}} \underset{h_n \sim P}{\mathbb{E}} e^{\frac{\lambda}{n}\Delta_n(h_n)} \tag{11}$$

Each $\Delta_i(h_i)$ is a bounded random variable with support in an interval of size 1. By Hoeffding's lemma we have

$$\underset{S_i}{\mathbb{E}} \underset{h_i \sim P}{\mathbb{E}} e^{\frac{\lambda}{n}\Delta_i(h_i)} \leq e^{\frac{\lambda^2}{8n^2m}}. \tag{12}$$

Therefore,

$$\underset{S_1,\dots,S_n}{\mathbb{E}} \underset{h_1 \sim P}{\mathbb{E}} \cdots \underset{h_n \sim P}{\mathbb{E}} \prod_{i=1}^{n} e^{\frac{\lambda}{n}\Delta_i(h_i)} \leq e^{\frac{\lambda^2}{8nm}}, \tag{13}$$

and by Markov's inequality, for any $\epsilon \in \mathbb{R}$,

$$\underset{S_1,\dots,S_n}{\mathbb{P}} \left( \underset{h_1 \sim P}{\mathbb{E}} \cdots \underset{h_n \sim P}{\mathbb{E}} \prod_{i=1}^{n} e^{\frac{\lambda}{n}\Delta_i(h_i)} \geq e^{\epsilon} \right) \leq e^{\frac{\lambda^2}{8nm} - \epsilon} \tag{14}$$

Hence

$$\underset{S_1,\dots,S_n}{\mathbb{P}} \left( \exists Q_1,\dots,Q_n : \frac{1}{n}\sum_{i=1}^{n} \Delta_i(Q_i) - \frac{1}{\lambda} D_{\mathrm{KL}}(Q_1 \times \cdots \times Q_n || P \times \dots \times P) \geq \frac{1}{\lambda}\epsilon \right) \leq e^{\frac{\lambda^2}{8nm} - \epsilon},$$

(15)

or equivalently,

$$\mathbb{P}_{S_1,\ldots,S_n}\Big(\forall Q_1,\ldots,Q_n : \frac{1}{n}\sum_{i=1}^{n}\Delta_i(Q_i) \tag{16}$$

$$\leq \frac{1}{\lambda}\sum_{i=1}^{n}D_{\mathrm{KL}}(Q_i\|P) + \frac{1}{\lambda}\log(\frac{2}{\delta}) + \frac{\lambda}{8nm}\Big) \geq 1 - \frac{\delta}{2}. \tag{17}$$

By applying a union bound for all the values of $\lambda \in \Lambda$ with $\Lambda = \{1,\ldots,4mn\}$ we get that

$$\mathbb{P}_{S_1,\ldots,S_n}\Big(\forall Q_1,\ldots,Q_n, \forall \lambda \in \Lambda : \frac{1}{n}\sum_{i=1}^{n}\Delta_i(Q_i) \tag{18}$$

$$\leq \frac{1}{\lambda}\sum_{i=1}^{n}D_{\mathrm{KL}}(Q_i\|P) + \frac{1}{\lambda}\log(\frac{8mn}{\delta}) + \frac{\lambda}{8nm}\Big) \geq 1 - \frac{\delta}{2} \tag{19}$$

Note that $\lfloor\lambda\rfloor \leq \lambda$ and $\frac{1}{\lfloor\lambda\rfloor} \leq \frac{1}{\lambda-1}$. Therefore, for all values of $\lambda \in (1,4mn)$ we have

$$\mathbb{P}_{S_1,\ldots,S_n}\Big(\forall Q_1,\ldots,Q_n, \forall \lambda \in (1,4mn) : \frac{1}{n}\sum_{i=1}^{n}\Delta_i(Q_i) \tag{20}$$

$$\leq \underbrace{\frac{1}{\lambda-1}\Big(\sum_{i=1}^{n}D_{\mathrm{KL}}(Q_i\|P) + \log(\frac{8mn}{\delta})\Big) + \frac{\lambda}{8mn}}_{=:\Gamma(\lambda)}\Big) \geq 1 - \frac{\delta}{2} \tag{21}$$

For any choice of $Q_1,\ldots,Q_n$, let $\lambda^* = \sqrt{8mn(\sum_{i=1}^{n}D_{\mathrm{KL}}(Q_i\|P) + \log(\frac{8mn}{\delta})) + 1}$. If $\lambda^* \geq 4mn$, that implies $\sum_{i=1}^{n}(D_{\mathrm{KL}}(Q_i\|P) + \log(\frac{8mn}{\delta}) + 1) \geq 2mn$ and $\Gamma(\lambda^*) > 1$, so Inequality (21) holds trivially. Otherwise, $\lambda^* \in (0,4mn)$, so Inequality (21) also holds. Therefore, we

$$\mathbb{P}_{S_1,\ldots,S_n}\Big(\forall Q_1,\ldots,Q_n, : \frac{1}{n}\sum_{i=1}^{n}\Delta_i(Q_i) \leq \sqrt{\frac{\sum_{i=1}^{n}D_{\mathrm{KL}}(Q_i\|P) + \log(\frac{8mn}{\delta}) + 1}{2mn}}\Big) \geq 1 - \frac{\delta}{2}. \tag{22}$$

In combination with (10), we obtain:

$$\mathbb{P}_{S_1,\ldots,S_n}\Big(\forall \mathcal{Q} : \widetilde{\mathrm{er}}(\mathcal{Q}) - \widehat{\mathrm{er}}(\mathcal{Q}) \leq \mathop{\mathbb{E}}_{A\sim\mathcal{Q}}\sqrt{\frac{\sum_{i=1}^{n}D_{\mathrm{KL}}(A(S_i)\|P) + \log(\frac{8mn}{\delta}) + 1}{2mn}}\Big) \geq 1 - \frac{\delta}{2}. \tag{23}$$

**Part II**   For upper bounding $\mathrm{er}(\mathcal{Q}) - \widetilde{\mathrm{er}}(\mathcal{Q})$ we have by a standard PAC-Bayesian bound (Maurer, 2004; Pérez-Ortiz et al., 2021),

$$\mathbb{P}_{S_1,\ldots,S_n}\Big(\forall \mathcal{Q} : \mathrm{er}(\mathcal{Q}) - \widetilde{\mathrm{er}}(\mathcal{Q}) \leq \sqrt{\frac{D_{\mathrm{KL}}(\mathcal{Q}\|\mathcal{P}) + \log(\frac{4\sqrt{n}}{\delta})}{2n}}\Big) \geq 1 - \frac{\delta}{2}. \tag{24}$$

**Combination**   We combine (23) and (24) by a union bound to obtain

$$\mathbb{P}_{S_1,\ldots,S_n}\Big(\forall \mathcal{Q} : \mathrm{er}(\mathcal{Q}) - \widehat{\mathrm{er}}(\mathcal{Q}) \leq \sqrt{\frac{D_{\mathrm{KL}}(\mathcal{Q}\|\mathcal{P}) + \log(\frac{4\sqrt{n}}{\delta})}{2n}} \tag{25}$$

$$+ \mathop{\mathbb{E}}_{A\sim\mathcal{Q}}\sqrt{\frac{\sum_{i=1}^{n}D_{\mathrm{KL}}(A(S_i)\|P) + \log(\frac{8mn}{\delta}) + 1}{2mn}}\Big) \geq 1 - \delta. \tag{26}$$

This concludes the proof of the theorem. □

**Discussion**   The difference in our proof and previous bounds in the similar settings (Pentina and Lampert, 2015; Rezazadeh, 2022) is in the upper bound for $\widetilde{\mathrm{er}}(\mathcal{Q}) - \widehat{\mathrm{er}}(\mathcal{Q})$ which is the average of

the generalization gaps for the $n$ training tasks that we observe. In our case we prove a generalization bound for arbitrary posteriors to be used by the tasks, using the change-of-measure inequality (CMI) for the transition from $Q_1 \times \cdots \times Q_n$ to $P \times \cdots \times P$. Because our bound holds uniformly in $Q_1, \ldots Q_n$, it also holds for the posteriors produced by a deterministic learning algorithm (a fixed $A$) or any distributions of algorithms (a hyperposterior $\mathcal{Q}$ over algorithms). This way we do not have to include the transition from $\mathcal{Q}$ to $\mathcal{P}$ in the CMI, as all prior work did. Consequently, in part of the proof we avoid getting a $KL(\mathcal{Q}||\mathcal{P})$ term which otherwise would be divided only by a $\frac{1}{m}$ factor, which is undesirable in the federated setting. Additionally compared to (Rezazadeh, 2022) we have better logarithmic dependency. They have a term of order $\frac{\log m \sqrt{n}}{m}$ which is not negligible in the case that we have large number of tasks and small number of samples per task, which is usual in the federated setting. In contrast, our logarithmic terms are divided by $n$.

**Proof of Theorem 3.2** We now instantiate the situation of Theorem 3.2 by specific choice of prior and posterior distributions. Let the learning algorithm be parameterized by the hypernetwork weights, $\eta_h$, and the embedding networks weights, $\eta_v$. As meta-posterior we use a Gaussian distribution, $\mathcal{Q} = \mathcal{Q}_h \times \mathcal{Q}_v$ for $\mathcal{Q}_h = \mathcal{N}(\eta_h; \alpha_h \mathrm{Id})$, and $\mathcal{Q}_v = \mathcal{N}(\eta_v; \alpha_v \mathrm{Id})$, where $\eta_h$ and $\eta_v$ are learnable and $\alpha_v$ and $\alpha_h$ are fixed. For any $(\bar{\eta}_h, \bar{\eta}_v) \sim \mathcal{Q}$ and training set $S$, the learning algorithm produces a posterior distribution $Q = \mathcal{N}(\theta; \alpha_\theta \mathrm{Id})$, where $\theta = h(v; \eta_h)$ with $v = \frac{1}{|S|} \sum_{(x,y)} \phi(x, y; \eta_v)$. As prior, we use $\mathcal{N}(0; \alpha_\theta \mathrm{Id})$. With these choices, we have $D_{\mathrm{KL}}(\mathcal{Q}, \mathcal{P}) = \alpha_h \|\eta_h\|^2 + \alpha_v |\eta_v\|^2$ and $D_{\mathrm{KL}}(Q_i, P) = \alpha_\theta \|\theta\|^2$. Inserting these into Theorem B.1, we recover Theorem 3.2.

# C APPENDIX - OPTIMIZATION

In this section, we formulate the technical assumptions for Theorem 3.1 and prove it. For the convenience of reading we repeat some notation from Section 3. For each client $i = 1, \ldots, n$ and parameter $\theta_i$, we note the client objective as $f_i(\theta_i)$ and $F_i(\eta) = f_i(h(S_i, \eta))$, where $h(S, \eta) = h(v(S, \eta_v), \eta_h)$ is a shorthand notation for the combination of hypernetwork and embedding network. The server objective we denote by $f(\eta)$, which in our setting consists only of regularization terms. The overall objective is $F(\eta) = \frac{1}{n} \sum_{i=1}^{n} F_i(\eta) + f(\eta)$. For the rest of this section, by $\eta_t$ we mean the parameters at iteration $t$, and $g_{t,i,j}$ is the stochastic gradient for client $i$ at global step $t$, local step $l$.

## C.1 ANALYTICAL ASSUMPTIONS

We make the following assumptions, which are common in the literature (Karimireddy et al., 2020; Fallah et al., 2020). Since our algorithm consists of multiple parts, we have to make the assumptions for different parts separately.

- Bounded Gradients: there exist constants, $b_1$, $b_2$, such that

$$\|\nabla_\theta f_i(\theta)\| \le b_1, \quad \|\nabla_\eta h(S, \eta)\| \le b_2 \tag{27}$$

  which, in particular, implies

$$\|h(S, \eta) - h(S, \eta')\| \le b_2 \|\eta - \eta'\| \tag{28}$$

- Smoothness: there exist constants, $L_1$, $L_2$, $L_3$, such that

$$\|\nabla_\theta f_i(\theta) - \nabla_\theta f_i(\theta')\| \le L_1 \|\theta - \theta'\| \tag{29}$$

$$\|\nabla_\eta h(S, \eta) - \nabla_\eta h(S, \eta')\| \le L_2 \|\eta - \eta'\| \tag{30}$$

$$\|\nabla_\eta f(\eta) - \nabla_\eta f(\eta')\| \le L_3 \|\eta - \eta'\| \tag{31}$$

- Bounded Dissimilarity: there exists a constant $\gamma_G$ with, such that with $\bar{F}(\eta) = \frac{1}{n} \sum_{i=1}^{n} F_i(\eta)$:

$$\frac{1}{n} \sum_{i=1}^{n} \mathbb{E}[\|\nabla F_i(\eta) - \nabla \bar{F}(\eta)\|^2] \le \gamma_G^2 \tag{32}$$

- Bounded Variance: there exists constants $\sigma_1$, $\sigma_2$, $\sigma_3$, such that, for batches $B_i \subset S_i$ of size $b < |S_i|$:

$$\mathbb{E} \|\nabla_\theta(f_i(\theta)) - \widetilde{\nabla}_\theta(f_i(\theta))\|^2 \le \sigma_1^2 \tag{33}$$

$$\mathbb{E} \|\nabla h(B_i, \eta) - \nabla h(S_i, \eta)\|^2 \le \frac{\sigma_2^2}{b} \tag{34}$$

$$\mathbb{E} \|h(B_i, \eta) - h(S_i, \eta)\|^2 \le \frac{\sigma_3^2}{b} \tag{35}$$

## C.2 CONVERGENCE

**Lemma C.1.** *There exists a constant $L$ which $\nabla F(\eta)$ is $L$-smooth.*

*Proof.* For each $F_i$ we have

$$\|\nabla_\eta F_i(\eta) - \nabla_\eta F_i(\eta')\| = \|\nabla_\theta f_i(h(S_i, \eta))\nabla_\eta h(S_i, \eta)^T - \nabla_\theta f_i(h(S_i, \eta'))\nabla_\eta h(S_i, \eta')^T\| \tag{36}$$

$$= \|\nabla_\theta f_i(h(S_i, \eta))\nabla_\eta h(S_i, \eta)^T - \nabla_\theta f_i(h(S_i, \eta))\nabla_\eta h(S_i, \eta')^T \tag{37}$$

$$+ \nabla_\theta f_i(h(S_i, \eta))\nabla_\eta h(S_i, \eta')^T - \nabla_\theta f_i(h(S_i, \eta'))\nabla_\eta h(S_i, \eta')^T\| \tag{38}$$

$$\le \|\nabla_\theta f_i(h(S_i, \eta))\nabla_\eta h(S_i, \eta)^T - \nabla_\theta f_i(h(S_i, \eta))\nabla_\eta h(S_i, \eta')^T\| \tag{39}$$

$$+ \|\nabla_\theta f_i(h(S_i, \eta))\nabla_\eta h(S_i, \eta')^T - \nabla_\theta f_i(h(S_i, \eta'))\nabla_\eta h(S_i, \eta')^T\| \tag{40}$$

$$\le b_1 L_2 \|\eta - \eta'\| + b_2 L_1 b_2 \|\eta - \eta'\| = (b_1 L_2 + b_2 L_1 b_2)\|\eta - \eta'\| = L'\|\eta - \eta'\| \tag{41}$$

Therefore for $F$ we have

$$\|\nabla_\eta F(\eta) - \nabla_\eta F(\eta')\| \le \|\frac{1}{n}\sum_{i=1}^n \nabla_\eta F_i(\eta) + \nabla_\eta f(\eta) - \frac{1}{n}\sum_{i=1}^n \nabla_\eta F_i(\eta') - \nabla_\eta f(\eta')\| \tag{42}$$

$$\le \frac{1}{n}\sum_{i=1}^n \|\nabla_\eta F_i(\eta) - \nabla_\eta F_i(\eta')\| + \|\nabla_\eta f(\eta) - \nabla_\eta f(\eta')\| \le (L' + L_3)\|\eta - \eta'\| = L\|\eta - \eta'\| \tag{43}$$

$$\square$$

**Lemma C.2.** *For independent random vectors $a_1, ..., a_n$ we have*

$$\mathbb{E}[\|\sum_{i \in C} a_i - \mathbb{E}[a_i]\|^2] \le \frac{c}{n}\sum_i \mathbb{E}[\|a_i - \mathbb{E}[a_i]\|^2] \tag{44}$$

*Also if $\mathbb{E}[a_i] \le a$ for each $i$, we have*

$$\mathbb{E}[\|\sum_{i \in C} a_i - \mathbb{E}[a_i]\|^2] \le \frac{c}{n}\sum_i \mathbb{E}[\|a_i - \mathbb{E}[a_i]\|^2] + c^2 a^2 \tag{45}$$

*Proof.* By expanding the power 2 we get

$$\mathbb{E}[\|\sum_{i \in C} a_i - \mathbb{E}[a_i]\|^2] = \mathbb{E}[\sum_{i \in C} \|a_i - \mathbb{E}[a_i]\|^2] + \mathbb{E}[\sum_{i \ne j \in C} \langle a_i - \mathbb{E}[a_i], a_j - \mathbb{E}[a_j]\rangle] \tag{46}$$

$$\le \frac{\binom{n-1}{c-1}}{\binom{n}{c}}\sum_i \mathbb{E}[\|a_i - \mathbb{E}[a_i]\|^2] + \frac{\binom{n-2}{c-2}}{\binom{n}{c}}\sum_{i \ne j} \mathbb{E}[\langle a_i - \mathbb{E}[a_i], a_j - \mathbb{E}[a_j]\rangle] \tag{47}$$

$$= \frac{\binom{n-1}{c-1}}{\binom{n}{c}}\sum_i \mathbb{E}[\|a_i - \mathbb{E}[a_i]\|^2] - \frac{2\binom{n-2}{c-2}}{\binom{n}{c}}\sum_i \mathbb{E}[\|a_i - \mathbb{E}[a_i]\|^2] \tag{48}$$

$$\le \frac{c}{n}\sum_i \mathbb{E}[\|a_i - \mathbb{E}[a_i]\|^2] \tag{49}$$

For the second part we have

$$\mathbb{E}[\|\sum_{i \in C} a_i\|^2] = \mathbb{E}[\|\sum_{i \in C} a_i - \mathbb{E}[a_i] + \mathbb{E}[a_i]\|^2] \tag{50}$$

$$= \mathbb{E}[\|\sum_{i \in C} a_i - \mathbb{E}[a_i]\|^2] + \mathbb{E}[\|\sum_{i \in C} \mathbb{E}[a_i]\|^2] + 2\,\mathbb{E}[\langle \sum_{i \in C} a_i - \mathbb{E}[a_i], \sum_{i \in C} \mathbb{E}[a_i]\rangle] \tag{51}$$

$$= \mathbb{E}[\|\sum_{i \in C} a_i - \mathbb{E}[a_i]\|^2] + \mathbb{E}[\|\sum_{i \in C} \mathbb{E}[a_i]\|^2] \leq \frac{c}{n} \sum_i \mathbb{E}[\|a_i - \mathbb{E}[a_i]\|^2] + c^2 a^2 \tag{52}$$

$\square$

**Lemma C.3.** *For any step $t$ and client $i$ the following inequality holds:*

$$\mathbb{E}\,\|\nabla F_i(\eta_t) - \nabla_\theta f_i(h(B_i, \eta_t) - \sum_{j=0}^{l-1} g_{t,i,j}(h(B_i, \eta_t)))\nabla h(B_i, \eta_t)^T\|^2 \tag{53}$$

$$\leq 27 L_1^2 b_2^2 \beta^2 l^2 (b_1^2 + \sigma_1^2) + 3b_1^2 \sigma_2^2 + 6L_1^2 b_2^2 \sigma_3^2 \tag{54}$$

*Proof.* By adding and subtracting some immediate terms we get

$$\mathbb{E}\,\|\nabla F_i(\eta_t) - \nabla_\theta f_i(h(B_i, \eta_t) - \sum_{j=0}^{l-1} g_{t,i,j}(h(B_i, \eta_t)))\nabla h(B_i, \eta_t)^T\|^2 \tag{55}$$

$$= \mathbb{E}\,\|\nabla F_i(\eta_t) - \nabla_\theta f_i(h(S_i, \eta_t) - \beta \sum_{j=0}^{l-1} g_{t,i,j}(h(S_i, \eta_t)))\nabla h(S_i, \eta_t)^T \tag{56}$$

$$+ \nabla_\theta f_i(h(S_i, \eta_t) - \beta \sum_{j=0}^{l-1} g_{t,i,j}(h(S_i, \eta_t)))(\nabla h(S_i, \eta_t)^T - \nabla h(B_i, \eta_t)^T) \tag{57}$$

$$+ (\nabla_\theta f_i(h(S_i, \eta_t) - \beta \sum_{j=0}^{l-1} g_{t,i,j}(h(S_i, \eta_t))) - \nabla_\theta f_i(h(B_i, \eta_t) - \beta \sum_{j=0}^{l-1} g_{t,i,j}(h(B_i, \eta_t))))\nabla h(B_i, \eta_t)^T\|^2$$

$$\tag{58}$$

And by Cauchy–Schwarz inequality we get

$$\mathbb{E} \, \| \nabla F_i(\eta_t) - \nabla_\theta f_i(h(B_i, \eta_t) - \sum_{j=0}^{l-1} g_{t,i,j}(h(B_i, \eta_t))) \nabla h(B_i, \eta_t)^T \|^2 \tag{59}$$

$$\leq 3 \, \mathbb{E} \, \| \nabla_\theta f_i(h(S_i, \eta_t)) \nabla h(S_i, \eta_t)^T - \nabla_\theta f_i(h(S_i, \eta_t) - \beta \sum_{j=0}^{l-1} g_{t,i,j}(h(S_i, \eta_t))) \nabla h(S_i, \eta_t)^T \|^2 \tag{60}$$

$$+ 3 \, \mathbb{E} \, \| \nabla_\theta f_i(h(S_i, \eta_t) - \beta \sum_{j=0}^{l-1} g_{t,i,j}(h(S_i, \eta_t)))(\nabla h(B_i, \eta_t)^T - \nabla h(S_i, \eta_t)^T) \|^2 \tag{61}$$

$$+ 3 \, \mathbb{E} \, \| (\nabla_\theta f_i(h(B_i, \eta_t) - \beta \sum_{j=0}^{l-1} g_{t,i,j}(h(B_i, \eta_t))) - \nabla_\theta f_i(h(S_i, \eta_t) - \beta \sum_{j=0}^{l-1} g_{t,i,j}(h(S_i, \eta_t)))) \nabla h(B_i, \eta_t)^T \|^2 \tag{62}$$

$$\leq 3 \, \mathbb{E}( \| \nabla_\theta f_i(h(S_i, \eta_t)) - \nabla_\theta f_i(h(S_i, \eta_t) - \beta \sum_{j=0}^{l-1} g_{t,i,j}(h(S_i, \eta_t))) \|^2 \| \nabla h(S_i, \eta_t) \|^2 ) \tag{63}$$

$$+ 3 \, \mathbb{E}( \| \nabla_\theta f_i(h(S_i, \eta_t) - \beta \sum_{j=0}^{l-1} g_{t,i,j}(h(S_i, \eta_t))) \|^2 \| \nabla h(B_i, \eta_t) - \nabla h(S_i, \eta_t) \|^2 ) \tag{64}$$

$$+ 3 \, \mathbb{E}( \| \nabla_\theta f_i(h(B_i, \eta_t) - \beta \sum_{j=0}^{l-1} g_{t,i,j}(h(B_i, \eta_t))) - \nabla_\theta f_i(h(S_i, \eta_t) - \beta \sum_{j=0}^{l-1} g_{t,i,j}(h(S_i, \eta_t))) \|^2 \| \nabla h(B_i, \eta_t) \|^2 ) \tag{65}$$

$$\leq 3 L_1^2 b_2^2 \beta^2 \, \mathbb{E} \, \| \sum_{j=0}^{l-1} g_{t,i,j}(h(S_i, \eta_t)) \|^2 + 3 b_1^2 \, \mathbb{E} \, \| \nabla h(B_i, \eta_t) - \nabla h(S_i, \eta_t) \|^2 \tag{66}$$

$$+ 6 L_1^2 b_2^2 \, \mathbb{E} \, \| h(B_i, \eta_t) - h(S_i, \eta_t) \|^2 + 6 L_1^2 b_2^2 \beta^2 \, \mathbb{E} \, \| \sum_{j=0}^{l-1} g_{t,i,j}(h(S_i, \eta_t)) - \sum_{j=0}^{l-1} g_{t,i,j}(h(B_i, \eta_t)) \|^2 \tag{67}$$

$$\leq 3 L_1^2 b_2^2 \beta^2 l^2 (b_1^2 + \sigma_1^2) + 3 b_1^2 \sigma_2^2 + 6 L_1^2 b_2^2 \sigma_3^2 + 24 L_1^2 b_2^2 \beta^2 l^2 (b_1^2 + \sigma_1^2) \tag{68}$$
$$= 27 L_1^2 b_2^2 \beta^2 l^2 (b_1^2 + \sigma_1^2) + 3 b_1^2 \sigma_2^2 + 6 L_1^2 b_2^2 \sigma_3^2 \tag{69}$$

which we used the assumptions on the bounded gradients and bounded variance and Lemma C.2. $\quad\square$

**Lemma C.4.** *For any step $t$ the following inequality holds:*

$$\mathbb{E}[\langle \nabla F(\eta_t), \eta_{t+1} - \eta_t \rangle] \leq \frac{-3\beta k}{4} \| \nabla F(\eta_t) \|^2 + 27 L_1^2 b_2^2 \beta^3 k^3 (b_1^2 + \sigma_1^2) + 3 \beta k b_1^2 \sigma_2^2 + 6 \beta k L_1^2 b_2^2 \sigma_3^2 \tag{70}$$

*Proof.*

$$\mathbb{E}_t[\langle \nabla F(\eta_t), \eta_{t+1} - \eta_t \rangle] = \mathbb{E}_t[\langle \nabla F(\eta_t), -\frac{\beta}{c} \sum_{i \in C} \sum_{l=1}^{k} g_{t,i,l}(h(B_i, \eta_t)) \nabla h(B_i, \eta_t)^T - \beta k \nabla f(\eta_t) \rangle] \tag{71}$$

$$= \mathbb{E}_t[\langle \nabla F(\eta_t), -\frac{\beta}{n} \sum_{i=1}^{n} \sum_{l=1}^{k} \nabla_\theta f_i(h(B_i, \eta_t) - \beta \sum_{j=0}^{l-1} g_{t,i,j}(h(B_i, \eta_t))) \nabla h(B_i, \eta_t)^T \rangle] \tag{72}$$

$$+ \mathbb{E}_t[\langle \nabla F(\eta_t), -\beta k \nabla f(\eta_t) \rangle] \tag{73}$$

$$= \frac{\beta}{n} \sum_{i=1}^{n} \sum_{l=1}^{k} \mathbb{E}_t[\langle \nabla F(\eta_t), \nabla F_i(\eta_t) - \nabla_\theta f_i(h(B_i, \eta_t) - \beta \sum_{j=0}^{l-1} g_{t,i,j}(h(B_i, \eta_t))) \nabla h(B_i, \eta_t)^T \rangle] \tag{74}$$

$$+ \frac{\beta}{n} \sum_{i=1}^{n} \sum_{l=1}^{k} \mathbb{E}_t[\langle \nabla F(\eta_t), -\nabla F_i(\eta_t) \rangle] + \mathbb{E}_t[\langle \nabla F(\eta_t), -\beta k \nabla f(\eta_t) \rangle] \tag{75}$$

By definition $F(\eta) = \frac{1}{n} \sum_{i=1}^{n} F_i(\eta) + f(\eta)$, and by Young's inequality and lemma C.3 we have

$$\mathbb{E}_t[\langle \nabla F(\eta_t), \eta_{t+1} - \eta_t \rangle] \tag{76}$$

$$\leq \frac{\beta k}{4} \|\nabla F(\eta_t)\|^2 + \frac{\beta}{n} \sum_{i=1}^{n} \sum_{l=1}^{k} \mathbb{E}_t \|\nabla F_i(\eta_t) - \nabla_\theta f_i(h(B_i, \eta_t) - \beta \sum_{j=0}^{l-1} g_{t,i,j}(h(B_i, \eta_t))) \nabla h(B_i, \eta_t)^T\|^2 \tag{77}$$

$$- \beta k \|\nabla F(\eta_t)\|^2 \tag{78}$$

$$\leq \frac{-3\beta k}{4} \|\nabla F(\eta_t)\|^2 + \frac{\beta}{n} \sum_{i=1}^{n} \sum_{l=1}^{k} (27 L_1^2 b_2^2 \beta^2 l^2 (b_1^2 + \sigma_1^2) + 3 b_1^2 \sigma_2^2 + 6 L_1^2 b_2^2 \sigma_3^2) \tag{79}$$

$$= \frac{-3\beta k}{4} \|\nabla F(\eta_t)\|^2 + 27 L_1^2 b_2^2 \beta^3 k^3 (b_1^2 + \sigma_1^2) + 3 \beta k b_1^2 \sigma_2^2 + 6 \beta k L_1^2 b_2^2 \sigma_3^2 \tag{80}$$

$$\square$$

**Lemma C.5.** *For any step $t$ the following inequality holds:*

$$\mathbb{E}[\|\eta_{t+1} - \eta_t\|^2] \leq \frac{4\beta^2 k}{c} \sigma_1^2 + \frac{4\beta^2 k^2}{c} \gamma_G^2 + 4\beta^2 k^2 \mathbb{E}[\|\nabla F(\eta_t)\|^2] \tag{81}$$

$$+ 108 L_1^2 b_2^2 \beta^4 k^4 (b_1^2 + \sigma_1^2) + 12 b_1^2 k^2 \beta^2 \sigma_2^2 + 24 L_1^2 b_2^2 k^2 \beta^2 \sigma_3^2 \tag{82}$$

*Proof.* The change of $\eta$ in one step of optimization is the average of the change caused by selected clients plus function $f(\eta)$ at the server.

$$\mathbb{E}[\|\eta_{t+1} - \eta_t\|^2] = \mathbb{E}[\| -\frac{\beta}{c} \sum_{i \in C} \sum_{l=1}^{k} g_{t,i,l}(h(B_i, \eta_t)) \nabla h(B_i, \eta_t)^T - \beta k \nabla f(\eta_t)\|^2] \tag{83}$$

Which we can write as

$$\mathbb{E}[\|\eta_{t+1} - \eta_t\|^2] = \frac{\beta^2}{c^2} \mathbb{E}[\|\sum_{i \in C} \sum_{l=1}^{k} g_{t,i,l}(h(B_i, \eta_t))\nabla h(B_i, \eta_t)^T + ck\nabla f(\eta_t)\|^2] \tag{84}$$

$$= \frac{\beta^2}{c^2} \mathbb{E}[\|\sum_{i \in C} \sum_{l=1}^{k} g_{t,i,l}(h(B_i, \eta_t))\nabla h(B_i, \eta_t)^T - \nabla_\theta f_i(h(B_i, \eta_t) - \sum_{j=0}^{l-1} g_{t,i,j}(h(B_i, \eta_t)))\nabla h(B_i, \eta_t)^T \tag{85}$$

$$+ \sum_{i \in C} \sum_{l=1}^{k} \nabla_\theta f_i(h(B_i, \eta_t) - \sum_{j=0}^{l-1} g_{t,i,j}(h(B_i, \eta_t)))\nabla h(B_i, \eta_t)^T - \nabla F_i(\eta_t) \tag{86}$$

$$+ \sum_{i \in C} \sum_{l=1}^{k} (\nabla F_i(\eta_t) + \nabla f(\eta_t) - \nabla F(\eta_t)) + \sum_{i \in C} \sum_{l=1}^{k} \nabla F(\eta_t)\|^2] \tag{87}$$

And by Cauchy–Schwarz inequality we get

$$\mathbb{E}[\|\eta_{t+1} - \eta_t\|^2] \tag{88}$$

$$\leq \frac{4\beta^2}{c^2} \mathbb{E}[\|\sum_{i \in C} \sum_{l=1}^{k} g_{t,i,l}(h(B_i, \eta_t))\nabla h(B_i, \eta_t)^T - \nabla_\theta f_i(h(B_i, \eta_t) - \sum_{j=0}^{l-1} g_{t,i,j}(h(B_i, \eta_t)))\nabla h(B_i, \eta_t)^T\|^2] \tag{89}$$

$$+ \frac{4\beta^2}{c^2} \mathbb{E}[\|\sum_{i \in C} \sum_{l=1}^{k} \nabla_\theta f_i(h(B_i, \eta_t) - \sum_{j=0}^{l-1} g_{t,i,j}(h(B_i, \eta_t)))\nabla h(B_i, \eta_t)^T - \nabla F_i(\eta_t)\|^2] \tag{90}$$

$$+ \frac{4\beta^2}{c^2} \mathbb{E}[\|\sum_{i \in C} \sum_{l=1}^{k} \nabla F_i(\eta_t) + \nabla f(\eta_t) - \nabla F(\eta_t)\|^2] + \frac{4\beta^2}{c^2} \mathbb{E}[\|\sum_{i \in C} \sum_{l=1}^{k} \nabla F(\eta_t)\|^2] \tag{91}$$

Note that $\nabla F(\eta) - \nabla f(\eta) = \frac{1}{n} \sum_{i=1}^{n} \nabla F_i(\eta)$. By lemma C.2, C.3, the bounded dissimilarity assumption and simplifying terms we get

$$\mathbb{E}[\|\eta_{t+1} - \eta_t\|^2] \tag{92}$$

$$\leq \frac{4\beta^2 k}{c} \sigma_1^2 + \frac{4\beta^2 k^2}{cn} \sum_i \mathbb{E}[\|\nabla F_i(\eta_t) - (\frac{1}{n} \sum_{i=1}^{n} \nabla F_i(\eta_t))\|^2] + 4\beta^2 k^2 \mathbb{E}[\|\nabla F(\eta_t)\|^2] \tag{93}$$

$$+ 108 L_1^2 b_2^2 \beta^4 k^4 (b_1^2 + \sigma_1^2) + 12 b_1^2 k^2 \beta^2 \sigma_2^2 + 24 L_1^2 b_2^2 k^2 \beta^2 \sigma_3^2 \tag{94}$$

$$\leq \frac{4\beta^2 k}{c} \sigma_1^2 + \frac{4\beta^2 k^2}{c} \gamma_G^2 + 4\beta^2 k^2 \mathbb{E}[\|\nabla F(\eta_t)\|^2] \tag{95}$$

$$+ 108 L_1^2 b_2^2 \beta^4 k^4 (b_1^2 + \sigma_1^2) + 12 b_1^2 k^2 \beta^2 \sigma_2^2 + 24 L_1^2 b_2^2 k^2 \beta^2 \sigma_3^2 \tag{96}$$

$$\square$$

We are now ready to prove Theorem 3.1, which we restate for the convenience of the reader.

**Theorem 3.1.** *Under the assumptions C.1 PeFLL's optimization after $T$ steps fulfills*

$$\frac{1}{T} \sum_{t=1}^{T} \mathbb{E}\|\nabla F(\eta_t)\|^2 \leq \frac{(F(\eta_0) - F_*)}{\sqrt{cT}} + \frac{224 c L_1^2 b_1^2 b_2^2}{T} + 8 b_1^2 \sigma_2^2 + 14 L_1^2 b_2^2 \sigma_3^2 \tag{97}$$

$$+ \frac{4L(2\sigma_1^2 + k\gamma_G^2)}{k\sqrt{cT}} \tag{98}$$

*Proof.* By smoothness of $F$ we have

$$\mathbb{E}[F(\eta_{t+1}) - F(\eta_t)] \leq \mathbb{E}\langle \nabla F(\eta_t), \eta_{t+1} - \eta_t \rangle + \frac{L}{2} \mathbb{E}\|\eta_{t+1} - \eta_t\|^2 \tag{99}$$

And by combining it with Lemma C.4 and C.5 and simplifying terms we get

$$\mathbb{E}[F(\eta_{t+1}) - F(\eta_t)] \le \mathbb{E}\langle \nabla F(\eta_t), \eta_{t+1} - \eta_t \rangle + \frac{L}{2} \mathbb{E} \|\eta_{t+1} - \eta_t\|^2 \tag{100}$$

$$\le \frac{-3\beta k}{4} \mathbb{E} \|\nabla F(\eta_t)\|^2 + 27L_1^2 b_2^2 \beta^3 k^3 (b_1^2 + \sigma_1^2) + 3\beta k b_1^2 \sigma_2^2 + 6\beta k L_1^2 b_2^2 \sigma_3^2 \tag{101}$$

$$+ \frac{2L\beta^2 k}{c} \sigma_1^2 + \frac{2L\beta^2 k^2}{c} \gamma_G^2 + 2L\beta^2 k^2 \mathbb{E}[\|\nabla F(\eta_t)\|^2] \tag{102}$$

$$+ 56LL_1^2 b_2^2 \beta^4 k^4 (b_1^2 + \sigma_1^2) + 6Lb_1^2 k^2 \beta^2 \sigma_2^2 + 12LL_1^2 b_2^2 k^2 \beta^2 \sigma_3^2 \tag{103}$$

$$= (\frac{-3\beta k}{4} + 2L\beta^2 k^2) \mathbb{E} \|\nabla F(\eta_t)\|^2 + (27L_1^2 b_2^2 \beta^3 k^3 + 56LL_1^2 b_2^2 \beta^4 k^4) b_1^2 \tag{104}$$

$$+ 3\beta k b_1^2 \sigma_2^2 + 6Lb_1^2 k^2 \beta^2 \sigma_2^2 + 6\beta k L_1^2 b_2^2 \sigma_3^2 + 12LL_1^2 b_2^2 k^2 \beta^2 \sigma_3^2 \tag{105}$$

$$+ (\frac{L\beta^2 k}{c} + 27L_1^2 b_2^2 \beta^3 k^3 + 56LL_1^2 b_2^2 \beta^4 k^4) \sigma_1^2 + \frac{L\beta^2 k^2}{c} \gamma_G^2 \tag{106}$$

$$\le \frac{-\beta k}{2} \mathbb{E} \|\nabla F(\eta_t)\|^2 + 28L_1^2 b_1^2 b_2^2 \beta^3 k^3 + 4\beta k b_1^2 \sigma_2^2 + 7\beta k L_1^2 b_2^2 \sigma_3^2 \tag{107}$$

$$+ \frac{3L\beta^2 k}{c} \sigma_1^2 + \frac{2L\beta^2 k^2}{c} \gamma_G^2 \tag{108}$$

Therefore

$$\mathbb{E}[F(\eta_{t+1}) - F(\eta_t)] \le \frac{-\beta k}{2} \mathbb{E} \|\nabla F(\eta_t)\|^2 + 28L_1^2 b_1^2 b_2^2 \beta^3 k^3 + 4\beta k b_1^2 \sigma_2^2 + 7\beta k L_1^2 b_2^2 \sigma_3^2 \tag{109}$$

$$+ \frac{3L\beta^2 k}{c} \sigma_1^2 + \frac{2L\beta^2 k^2}{c} \gamma_G^2 \tag{110}$$

Hence

$$\mathbb{E} \|\nabla F(\eta_t)\|^2 \le \frac{2}{\beta k} \mathbb{E}[F(\eta_t) - F(\eta_{t+1})] + 56L_1^2 b_1^2 b_2^2 \beta^2 k^2 + 8b_1^2 \sigma_2^2 + 14L_1^2 b_2^2 \sigma_3^2 \tag{111}$$

$$+ \frac{6L\beta}{c} \sigma_1^2 + \frac{4L\beta k}{c} \gamma_G^2 \tag{112}$$

And

$$\frac{1}{T} \sum_{t=1}^{T} \mathbb{E} \|\nabla F(\eta_t)\|^2 \le \frac{2(F(\eta_0) - F_*)}{\beta k T} + 56L_1^2 b_1^2 b_2^2 \beta^2 k^2 + 8b_1^2 \sigma_2^2 + 14L_1^2 b_2^2 \sigma_3^2 \tag{113}$$

$$+ \frac{6L\beta}{c} \sigma_1^2 + \frac{4L\beta k}{c} \gamma_G^2 \tag{114}$$

by setting $\beta = \frac{2\sqrt{c}}{k\sqrt{T}}$ we get

$$\frac{1}{T} \sum_{t=1}^{T} \mathbb{E} \|\nabla F(\eta_t)\|^2 \le \frac{(F(\eta_0) - F_*)}{\sqrt{cT}} + \frac{224cL_1^2 b_1^2 b_2^2}{T} + 8b_1^2 \sigma_2^2 + 14L_1^2 b_2^2 \sigma_3^2 \tag{115}$$

$$+ \frac{L(6\sigma_1^2 + 4k\gamma_G^2)}{k\sqrt{cT}} \tag{116}$$

$$\square$$

