# OpenReview forum: "PeFLL: Personalized Federated Learning by Learning to Learn"
_ICLR.cc/2024/Conference — ICLR 2024 poster_

### Official Review · Reviewer_tVNw · 2023-10-29

**Soundness:** 3 good
**Presentation:** 3 good
**Contribution:** 3 good
**Rating:** 6
**Confidence:** 4

**Summary:**

This paper proposes a learning-to-learn framework for personalized FL, where the goal is to learn a global hypernetwork that outputs each clients' personalized model. A shared embedding network is also trained, which produces each client's descriptor based on its local dataset. During inference, any client (even clients that did not participate during training) can send its descriptor to the server and receive the personalized model. This enables each client to obtain the model without further finetuning at test-time. Convergence analysis as well as generalization bound analysis are provided. The effectiveness of proposed idea is also confirmed via experiments.

**Strengths:**

1. The idea of taking advantage of learning-to-learn and hypernetworks in personalized FL is interesting.

2. The proposed idea has several advantages. Specifically, any client can obtain its personalized model by conducting only forward propagations.

3. Experimental results show significant performance advantage of the proposed method.

4. Theoretical results are also provided, further strengthening the paper.

**Weaknesses:**

1. All experiments are conducted using LeNet-style models, which are relatively outdated. I would like to see the performance on ResNet-style models.

2. In Table 2, the performance of "Local" is missing. Moreover, during experiments, what is the number of local updates for the baselines like Per-FedAvg and pFedMe. How do their performance improve as the number of local updates increases, and can they achieve similar performance compared to the proposed PeFLL?

**Questions:**

See weakness above.




------------------------------------------------

Final Update: I appreciate the authors for providing additional experiments. I think this paper is above the acceptance threshold, and would like to keep my original score 6.

---

> ### Author Response · Authors · 2023-11-17
>
> Thank you for your review, we answer your specific concerns below.
>
> > All experiments are conducted using LeNet-style models, which are relatively outdated. I would like to see the performance on ResNet-style models.
>
> Please see our reply to the first question of reviewer [Couf](https://openreview.net/forum?id=MrYiwlDRQO&noteId=zjBDwmhek) about the architecture and the size of the network.
>
>
> > In Table 2, the performance of "Local" is missing.
>
> We can include this. It is the same procedure as in Table 1 operating on the same kind of data, so the values for local in Table 2 will not differ substantially from those in Table 1.
>
>
> > Moreover, during experiments, what is the number of local updates for the baselines like Per-FedAvg and pFedMe. How do their performance improve as the number of local updates increases, and can they achieve similar performance compared to the proposed PeFLL?
>
> We assume by ‘local updates’ here you mean the number update steps used to personalize/finetune to a test client to obtain the results quoted in Table 2. The personalization on each test client was simply run locally until convergence, which does not usually require many steps due to the small amount of data on each client. The reported result is based on the best validation step. Therefore, we do not see that the results for Per-FedAvg/pFedMe could improve by even more steps.

---

> > ### Comment · Reviewer_tVNw · 2023-11-18
> > **Thanks**
> >
> > Thanks for the response. However, I do not agree with the authors that experiments with the current model is enough. Although FL can be applied to low-power devices, it could be also applied to other applications like autonomous vehicles, hospitals where clients have enough powers, and in these cases, one might use larger models. Conducting additional experiments on ResNet does not provide any negatives to the paper, but actually will validate the applicability of the proposed method. I still feel that only using LeNet-style model is not enough.

---

### Official Review · Reviewer_Fw5z · 2023-10-30

**Soundness:** 3 good
**Presentation:** 4 excellent
**Contribution:** 4 excellent
**Rating:** 8
**Confidence:** 4

**Summary:**

The authors present a federated training algorithm which allows clients to receive a personalized model at test-time, based on their individual local dataset. This is achieved by predicting personalized network parameters based on an individual client's descriptor. This descriptor (a vector embedding), is generated by passing the local dataset through a set network. The embedding computation is done locally, while the (expensive) hypernetwork is run on the server.
The training algorithm involves reversing the forward-pass and computing partial gradients at the server and the individual clients.

-- post rebuttal --
The authors have addressed my concerns and I do recommend acceptance.
While some question with respect to the method's generalizability to new tasks & models remain, I believe it to be a worthwhile method to experiment with for a new personalization use-case.

**Strengths:**

This paper addresses a highly relevant problem in an innovative way. The proposed test-time generation of personalized models through a server-side hypernetwork is original and well-described. The paper reads very well, I enjoyed following the authors' red line through the paper.

**Weaknesses:**

The weaknesses of this paper revolve around the experimental evaluation, specifically the base-lines that the authors consider given their choice of the experimental setting. The evaluation pipeline raises som questions, detailed below.
The comparison to baselines is somewhat unfair, due to a lack of measuring communication overhead.
Finally, i believe the discussion of related works should include the topic of Split-Learning, a version of FL where parts of the forward-pass are performed across different devices (e.g. client and server). PeFFL is an example of split-learning under that definition.

**Questions:**

- The authors only consider the label-skew setting of object-classification computer vision problems C10/C100 and Femnist. Specifically for C10/C100, the authors assign two labels per client. The provided code (mask_absent is True by default) suggest that their method consequently reduces to a binary classification problem at test time. Can you confirm this choice of evaluation is consistent with all baselines? Evaluation in line 374 & 375 of train.py concerns the test-clients. Following into line 20, evaluation is done. The default args embed_split=True suggest that you are using the training-set to compute embeddings, as well as (lines 58-76) using it to test for presence of classes on that client. It seems that for all (train & test) clients, you assume the presence of a labelled training set, which allows for masking your logits. Lines 24-38 suggest you're treating training & test clients identically. I would expect a "test-client" to not have any labelled data available, meaning that the logit-masking should not be allowed. Can you clarify your evaluation procedure for me? Do "test-clients" use a different training-set for embedding computation? I would expect those to be identical.
For the training-clients, I would be curious to understand if you use the labelled training data for embedding computation, or throw those away for the validation-set in Table 1a. You claim that the client descriptor network is not susceptible to overfitting. I would ask you to show evidence for that claim.
- How does PeFLL fair in non-label skew settings? E.g. I would recommend the Shakespeare dataset (https://arxiv.org/abs/1812.01097)
- In order to fairly compare PeFFL against baselines, I believe it is necessary to introduce the communication overhead that is introduced by the different methods. You argue that "It has low latency and communication costs", which should be clarified to be true only at test-time (and even then is higher than alternatives that function offline, such as FedAvg). During training, PeFFL is constrained to "FedSGD", i.e. a client cannot amortize communication with additional computation. Instead, forward-backward pass requires communication of various elements: Embedding, client-specific model and the gradients corresponding to both. I would like to see two analyses:
1.) GB of communication on the x-axis and test-client accuracy at the y-axis. This allows to draw iso-lines and understand the impact of the high-frequency communication of PeFFL.
2.) Introduction of another baseline, in which a larger, more flexible model is trained through FedAdam (https://arxiv.org/abs/2003.00295) with properly tuned hyper-parameters. The hypothesis I would like to see tested is if a more flexible model is able to generalize across all train & test clients without requiring personalization. For a given communication budget, should I choose PeFFL or can I fall back to a larger FedAvg-trained model?
- Another interesting experiment (though not required), would be to see a combination of classical FedAvg and PeFFL. Consider applying PeFFL only to the last layer(s) of a model, while training the rest through standard FedAvg. For object classification, it is usually the last layers that require personalization, while the initial layers can share jointly relevant features. Such a setup would allow the embedding network to be a separate head of the FedAvg-trained network.

Should the authors address my questions, I will consider raising my score

---

> ### Author Response · Authors · 2023-11-17
>
> Thank you for your detailed feedback, we address your concerns below.
>
> > I believe the discussion of related works should include the topic of Split-Learning
>
> Thank you for bringing this to our attention. We agree that it is both relevant and highly interesting. We will add this topic to our related work section and also include citations to [1,2]. Please let us know if you have other suggested related work. However, we would like to emphasize that we are not doing split learning exactly as described because we are not communicating gradients, but rather model differences after multiple updates (exactly as FedAvg does). We discuss this in more detail below.
>
> [1] Otkrist Gupta, Ramesh Raskar. Distributed learning of deep neural network over multiple agents, Journal of Network and Computer Applications, 2018
>
> [2] Chandra Thapa, M.A.P. Chamikara, Seyit Camtepe, Lichao Sun. SplitFed: When Federated Learning Meets Split Learning, AAAI, 2022
>
>
> > The authors only consider the label-skew setting of object-classification computer vision problems C10/C100 and FEMNIST. Specifically for C10/C100, the authors assign two labels per client. The provided code (mask_absent is True by default) suggests that their method consequently reduces to a binary classification problem at test time. Can you confirm this choice of evaluation is consistent with all baselines?
>
> First, what you describe is the setting for CIFAR10. For CIFAR100, each client actually has 10 classes, and for FEMNIST they can have all classes.
> Second, for all baselines the clients implicitly or explicitly use information about their subset of relevant classes. For example, kNN-Per performs kNN, so it can never predict a class not present in the client’s labeled training set. All other personalized methods train or finetune a classifier locally on a labeled training set, even for the test clients, so classes not present in their training data are quickly suppressed.
>
>
> > It seems that for all (train & test) clients, you assume the presence of a labeled training set. [...] I would expect a "test-client" to not have any labelled data available.
>
> Indeed, in our setting “test clients” differ from “train clients” only by the fact that they appear later, after the server is done with training. Maybe a better phrasing would be “clients seen during the server’s training phase” and “clients seen after the server’s training phase is finished”.
>
> In the supervised setting we study, labels are certainly needed during the training phase, and therefore it makes sense to assume that clients occurring later also have them. Indeed, none of the baselines to which we compare would be able to personalize to a test client that has no labeled data.
>
> However, we find the suggestion of learning personalized models for clients without labels for their data also quite exciting. Specifically, all baselines we report, except plain FedAvg, would fail to produce any classifiers for new clients in that setting. PeFLL, however, could work simply by using an embedding network that operates on the input data without labels. We are now running experiments in this setting (and without masking). We will report it as soon as it finishes, and we thank the reviewer for the suggestion.
>
> > Can you clarify your evaluation procedure for me? Do "test-clients" use a different training-set for embedding computation? I would expect those to be identical. For the training-clients, I would be curious to understand if you use the labelled training data for embedding computation, or throw those away for the validation-set in Table 1a.
>
> Train and test clients behave identically in that they receive the embedding network, compute their client descriptor from a batch of their data, send the descriptor back to the server and receive a personalized model. The only difference is that “train clients” subsequently participate in the server’s training step by performing gradient computations and sending parameter updates to the server.
>
>
>
> > You claim that the client descriptor network is not susceptible to overfitting. I would ask you to show evidence for that claim.
>
> Please see the paragraph in the experiment section **Analysis of Client Descriptors** as we believe this answers your question. In this experiment we explicitly test if the client descriptor (embedding) network is able to embed the test clients into descriptor space in a way that is consistent with the embeddings of the train clients. As the experiment shows (Figure 3), the embedding network does indeed learn to do this. If the embedding network simply overfit to the train clients then the test client descriptors would be meaningless.

---

> ### Author Response · Authors · 2023-11-17
>
> > How does PeFLL fare in non-label skew settings? E.g. I would recommend the Shakespeare dataset
>
> Note that FEMNIST is not a purely a label skew dataset, as different clients correspond to different writers. Thank you for suggesting the Shakespeare dataset, we will try to get experiments running and finished before the end of the discussion period.
>
>
> > During training, PeFFL is constrained to "FedSGD", i.e. a client cannot amortize communication with additional computation. Instead, forward-backward pass requires communication of various elements: Embedding, client-specific model and the gradients corresponding to both.
>
> This is not correct and we believe this could be leading to a **fundamental** misunderstanding of the communication costs of our method. We are **not** constrained to FedSGD. During training each client runs k local updates to their personalized model (line 10 of Algorithm 2). In our experiments k=50, while FedSGD would be k=1. The term that gets sent to the server to be ‘backpropped’ through the network is the difference between the initial personalized model and the final one **after** these 50 steps, (line 11 of Algorithm 2). We also refer you to our convergence results in Theorem 3.1, which shows that the number of local steps has the same effect as it has in FedAvg, and **proves that our procedure has equivalent convergence to FedAvg with k local steps per round** and not FedSGD. Since our convergence bound is in the same order of FedAvg, the number of global steps required for the convergence is the same as FedAvg, and we do not require more communication rounds than FedAvg.
>
>
> > Introduction of another baseline, in which a larger, more flexible model is trained through FedAdam (https://arxiv.org/abs/2003.00295) with properly tuned hyper-parameters. The hypothesis I would like to see tested is if a more flexible model is able to generalize across all train & test clients without requiring personalization. For a given communication budget, should I choose PeFFL or can I fall back to a larger FedAvg-trained model?
>
> As we mentioned for the previous question, we have the same number of communication rounds as FedAvg for convergence, just in each round we send two (small) networks instead of one. Also, our optimization is in fact already done using Adam.
>
> Regarding a larger model: we do agree that a large enough global model might also achieve high accuracy in the setting of our experiments. Note, however, that such a baseline would not be a fair comparison in terms of the computational load on client devices (which often have very limited compute and memory)
> In light of this fact and given our reply to your prior concern about the PeFLL’s communication cost, do you still think such a baseline is necessary?
>
>
> > Another interesting experiment (though not required), would be to see a combination of classical FedAvg and PeFFL. Consider applying PeFFL only to the last layer(s) of a model, while training the rest through standard FedAvg. For object classification, it is usually the last layers that require personalization, while the initial layers can share jointly relevant features. Such a setup would allow the embedding network to be a separate head of the FedAvg-trained network.
>
> We agree that this could be an interesting experiment, in particular as some early layers of the network could double as the embedding network. PeFFL would allow for such a setting without major modifications. Thank you for the suggestion!

---

> ### Author Response · Authors · 2023-11-18
> **Results of training without using labels for embedding:**
>
> We have now finished running the results for the experiments where the test clients do not possess labeled data. Specifically what we do is: the PeFLL embedding network is trained on the training clients using only the image data and no labels (embed_y = False in the code). The test clients then obtain their personalized model using only their unlabeled data and evaluate this model. Of course since they do not have labeled data there is no class masking (mask_absent = False in the code). We report below the results on the test clients. Note that the only baseline that is capable of addressing this setting is FedAvg, since all personalized FL methods require labeled data to be available at the test clients in order to obtain a personalized model.
>
>
> |              |     |      CIFAR10      |            |    |      CIFAR100       |            | FEMNIST    |
> |--------------|:------------:|:------------:|:------------:|:------------:|:-------------:|:------------:|:------------:|
> | #trn clients | 90         | 450        | 900        | 90         | 450         | 900        | 3237       |
> | FedAvg       | 48.6 ± 0.3 | 49.6 ± 1.0 | 51.7 ± 0.7 | 17.1 ± 0.9 |  19.3 ± 2.0 | 20.5 ± 1.0 | 81.9 ± 0.4 |
> | PeFLL        | 84.5 ± 0.6 | 83.6 ± 1.0 | 81.5 ± 0.7 | 26.3 ± 0.4 | 28.2 ± 0.6  | 30.3 ± 0.6 | 91.4 ± 0.1 |

---

> > ### Comment · Reviewer_Fw5z · 2023-11-21
> > **Thanks for the clarifications, additional work remains**
> >
> > I thank the authors for their clarifying responses to my questions.
> >
> > If I understand correctly, your additional results were run without label-masking and without including the labels during the training and evaluation of the embedding network.
> > With respect to label-masking, I am still lacking an explanation if this strategy is used during evaluation of all baselines. E.g. during evaluation of the FedAvg baseline. For the additional experiments (where you claim no label masking) you seem to have included the same numbers as in the main paper. Please clarify.
> >
> > I don't belive there is a need to rename or re-define train vs. test clients. The only necessary clarification concerns weather test-clients are assumed to have labels available, which you have answered to be the case.
> >
> > I believe indeed that the unsupervised personalization angle to be a strong argument for your paper and would encourage to lean into it more.
> >
> > The answer to your question regarding the use of training or evaluation data for the embedding computation is still not clear to me. For a training client, i.e. Table 1a, you assume the existence of a training set, which is used during training, as well as the existance of a test-set on that same client. For the computation of the client-embedding at evaluation-time, which dataset are you drawing a mini-batch from? The training-set (which was used to train the embedding network) or the test-set (which has not been "seen" by the training of the embedding network)?
> >
> > Thanks for pointing out the "Analysis of Client Descriptors"; Indeed, it answeres my question about the overfitting claim - sorry I didn't associate the two.
> >
> > Indeed your method goes beyond FedSGD for training of the $\theta_i$ per client. I was referring to $\delta \nu_v^{(i)}$ which is the result of a single forward-backward pass. Fundamentally, my question about the empirical comparisson wrt. communication budget has thus not been answered yet. At each communication round, the communication costs per client for FedAvg are $2*|\theta|$, i.e. twice the model size. For PeFLL, the communication costs are: $2*|\nu_v| + 2*|v| + 2*|\theta|$; These correspond to lines 3, 7, 9, 11, 13, 15 of your algorithm. So the question remains: If I spent the additional communication budget of PeFLL on a larger model for FedAvg, would that change the story? Equivalently, for a given communication budget, i.e. if I stopped training PeFLL earlier than a FedAvg training run with the same model $\theta$ in both experiments, which model would be better? Assuming you have retained logs for your existing training runs, it should be easy to create learning curves with GB at the x-axis and test-accuracy at the y-axis.
> >
> > You are asking if experiments with a larger model are still necessary. I'd argue that the plot I asked for would be enough for me. Given the other reviewer's ask for a larger model, it might be a a good idea regardless.
> >
> > For the mean-time, I will raise my score slightly, but I would need the following for further increase:
> > - Learning-curve with Accuracy vs. GB and discussion of those
> > - Shakespeare (or other dataset) results
> > - Clarification for label-masking across all baselines.
> > - Discussion of your additional results: Please take apart the effect of label-masking from the effect of adding labels to the embedding network. I am especially curious to understand why C100 degraded significantly: Is the embedding network relying on labels to perform well here? Is the label-masking relevant for C100 to work well in your setup?
> > - An updated version of the paper, where these results, comments and clarifications have been added.
> >
> > Again, thanks for the interesting work - I appreciate the ideas and contributions. The execution & evaluation is what I am critiquing here.

---

> > > ### Author Response · Authors · 2023-11-22
> > >
> > > Thank you for the constructive interaction. We try to address all of your concerns below.
> > >
> > >
> > > > If I understand correctly, your additional results were run without label-masking and without including the labels during the training and evaluation of the embedding network.
> > >
> > >
> > > That is correct. For these experiments, the labels were only used during the training phase to compute the on-client loss, but not for computing the client-embeddings. The clients used the networks produced by the hypernetwork in unmodified form, in particular without label masking.
> > >
> > >
> > > > With respect to label-masking, I am still lacking an explanation if this strategy is used during evaluation of all baselines. E.g. during evaluation of the FedAvg baseline. For the additional experiments (where you claim no label masking) you seem to have included the same numbers as in the main paper. Please clarify.
> > >
> > >
> > > Sorry if our previous answer was not clear enough. None of the other methods use explicit label masking during evaluation. We included label masking as part of the evaluation protocol for PeFLL because we believe this made for a more accurate comparison to the other personalized baselines which essentially do label masking when they personalize to new clients (e.g. kNN-Per only predicts labels in the training set, and other methods finetune on training data which quickly suppresses absent labels). However, we understand now that this could be seen as unfair and that strictly the same evaluation protocol should be used for all methods. To ensure this is the case we have now simply removed the label masking altogether from the main body of the text, and updated the results in Table 1 for PeFLL to be without masking.
> > >
> > >
> > > Our CIFAR10 results are almost not affected. The CIFAR100 results get worse, but still better than all baselines, except in one case. FEMNIST had not been using label masking anyway, because the clients are given externally and not created by subsampling labels in the first place.
> > >
> > >
> > > > I believe indeed that the unsupervised personalization angle to be a strong argument for your paper and would encourage to lean into it more.
> > >
> > >
> > > Thank you. We have included this situation and its results in the main section of the manuscript (at the expense of having to move the discussion of client extrapolation to the supplementary material)
> > >
> > >
> > > > The answer to your question regarding the use of training or evaluation data for the embedding computation is still not clear to me. For a training client, i.e. Table 1a, you assume the existence of a training set, which is used during training, as well as the existance of a test-set on that same client. For the computation of the client-embedding at evaluation-time, which dataset are you drawing a mini-batch from? The training-set (which was used to train the embedding network) or the test-set (which has not been "seen" by the training of the embedding network)?
> > >
> > >
> > > The client-embeddings are always computed on the clients’ training data, both during training and evaluation. In the code, the corresponding embed_split option is always set to train. The embedding vector’s role is, after all, to serve as a descriptor for a training set.
> > > Using the test data to compute this (and generate the personalized model) would be akin to training on the test data. We only use the test data to evaluate the quality of the personalized models.
> > >
> > >
> > > Note, however, that during training the train clients only use a single batch of their training data to compute their embedding vector. Hence, it is not the case that a client is always generating the same embedding vector across multiple rounds during training.

---

> > > > ### Author Response · Authors · 2023-11-22
> > > >
> > > > > Fundamentally, my question about the empirical comparisson wrt. communication budget has thus not been answered yet. [...] So the question remains: If I spent the additional communication budget of PeFLL on a larger model for FedAvg, would that change the story? Equivalently, for a given communication budget, i.e. if I stopped training PeFLL earlier than a FedAvg training run with the same model
> > > >
> > > >
> > > > Thank you for clarifying your concern. Indeed, the communication cost of PeFLL per round is approximately twice that of FedAvg ($\nu_v$ and $\theta$ are of comparable or same size, the size of $v$ is just a few dozen bytes).
> > > >
> > > >
> > > > As requested we have now included training curves that compare accuracy obtained for a given communication budget (measured in GB of transferred information). As we can see from the curves for any given communication budget PeFLL performs substantially better than FedAvg.
> > > >
> > > >
> > > > We do believe the comparison with respect to rounds also has merits, though, as this better reflects other aspects of federated learning training, such as recruiting clients to participate, client computation and client stragglers, which can be bottlenecks in practice.
> > > >
> > > >
> > > > > Please take apart the effect of label-masking from the effect of adding labels to the embedding network. I am especially curious to understand why C100 degraded significantly: Is the embedding network relying on labels to perform well here? Is the label-masking relevant for C100 to work well in your setup?
> > > >
> > > >
> > > > As can be seen from comparing the results for PeFLL in Table 6 (with masking, with label embedding), Table 1 (without masking, with label embedding) and Table 2 (without masking, without label embedding) on the Test Clients, for the case of CIFAR100, we experience drops in accuracies both when we remove masking and when we remove access to the labels for the embedding network. So for CIFAR100 it seems safe to conclude that both label masking and label embeddings were beneficial to increase performance, and the contributed to comparable amounts. For CIFAR10, however, label masking had little to no effect, while removing access to labels for embedding did lead to a small reduction in performance.
> > > >
> > > >
> > > > To summarize, we have modified our manuscript and uploaded the new version to OpenReview with all changes highlighted in blue. Our changes include the following aspects:
> > > >
> > > > 1) Learning-curves with Accuracy vs. GB, and discussion of these curves. These can be found in Appendix A.3, **Comparison of communication and computation cost with FedAvg**
> > > >
> > > >
> > > > 2) Results on the Shakespeare dataset, these can be found in Appendix A.3, **Shakespeare Dataset**
> > > >
> > > > We have not had time to run baselines other than FedAvg on this data before the end of the discussion period. This would have required larger code changes as none of the personalized FL baselines had done this in their works. We will do this afterwards, though, for a potential camera-ready version.
> > > >
> > > >
> > > > 3) We have removed label-masking from the main manuscript and updated the results in Table 1 to those of PeFLL without label masking. We include a discussion of label masking in the supplemental material, discussing it only as an extended scenario. See Appendix A.3, **Integrating known client label sets**.
> > > >
> > > >
> > > > We hope that this has addressed your remaining concerns. We were excited to see that you appreciate the technical contribution and we hope you will support our work in the rest of the process.

---

> > > > > ### Comment · Reviewer_Fw5z · 2023-11-23
> > > > > **Thanks for all the work**
> > > > >
> > > > > I appreciate the additional results!
> > > > > My asks were:
> > > > > - Learning-curve with Accuracy vs. GB and discussion of those
> > > > > - Shakespeare (or other dataset) results
> > > > > - Clarification for label-masking across all baselines.
> > > > > - Discussion of your additional results: Please take apart the effect of label-masking from the effect of adding labels to the embedding network. I am especially curious to understand why C100 degraded significantly: Is the embedding network relying on labels to perform well here? Is the label-masking relevant for C100 to work well in your setup?
> > > > > - An updated version of the paper, where these results, comments and clarifications have been added.
> > > > > As far as I can see, you delivered on all of those.
> > > > > The learning-curves are clearly showing two things:
> > > > > - Your method outperforms FedAvg strongly. Please add a larger model FedAvg baseline for the final version of the paper, i.e. increase your LeNet model to reasonable width such that you have twice the amount of parameters.
> > > > > - Your method seems to not generalize equally well to the LSTM & Shakespeare as it works for C10/C100. For the final version, I'd recommend looking into why that is.
> > > > >
> > > > > Considering your C100 results with Shakespeare, I believe that your method might not be as easily generalizable, i.e. I cannot expect it to "just work" and provide clear benefits when applying to a real-world scenario or any new scenario that goes beyond small-scale object classification. However I believe it to still be a valuable tool to experiment with and try out for a personalization scenario.
> > > > >
> > > > > I have updated my rating accordingly!

---

### Official Review · Reviewer_Couf · 2023-10-31

**Soundness:** 3 good
**Presentation:** 3 good
**Contribution:** 3 good
**Rating:** 5
**Confidence:** 3

**Summary:**

This paper proposes a learning-to-learn approach to personalized federated learning. A hypernetwork residing on the server outputs the parameters of the personalized models given “descriptors” of the clients. The descriptor is generated by an embedding model that is shared by the server to the clients so that they can use it on their private data. This method avoids some issues in personalized federated learning by construction, such as the need to fine-tune, how to generalize to unseen clients, clients having little data, system/data/model heterogeneity, etc. Convergence proofs and generalization bounds are given for their method PeFLL. Experiments show that this model is indeed effective by a large margin, especially at generalizing to unseen clients.

**Strengths:**

- The paper is well-written and the approach is interesting.
- Heterogeneity issues can be alleviated with a learning-to-learn approach.
- Generalizing to unseen clients makes sense with the embedding model. We intuitively assume that client similarity is dictated by the similarity of their data. Thus, clients with similar embeddings should have similar personalized models.
- Algorithms are well detailed and the code is shared in the supplementary materials for reproducibility.
- Experiments show significant improvements on FEMNIST and CIFAR-10/100, especially on clients unseen during training.
- The benefits of client embedding network is justified experimentally.

**Weaknesses:**

- I think that the experiments are not sufficient to conclude that PeFLL is superior. You have to demonstrate its improvement on architectures more modern than LeNet as well. It would also be great to report performance on federated datasets other than FEMNIST (e.g. other LEAF datasets).
- The computational efficiency of the model is questionable given that the hypernetwork is at least a 100 times bigger than the client's model, and this is only the case for these particular experiments. It might require a wider net to generate larger personalized models. In other words, LeNet on MNIST and CIFAR-10/100 might only need 100 "ranks" to perform well, whereas on more complicated models and datasets, a wider net would be needed for good performance, which makes scalability an issue. That is why I think experiments covering these settings are needed to conclude the practical benefits of PeFLL.

**Questions:**

- Can you share some details about training time, memory consumption, and computational efficiency of PeFLL vs. regular models?

---

> ### Author Response · Authors · 2023-11-17
>
> Thank you for your review, we address your comments below.
>
> > I think that the experiments are not sufficient to conclude that PeFLL is superior. You have to demonstrate its improvement on architectures more modern than LeNet as well.
>
> Note that the model we are using is not the classic “LeNet-5” with tanh-activation etc. that you might have in mind. It is a modern ConvNet inspired by the LeNet architecture but bigger and with ReLU activations. We use this architecture to allow a fair comparison of PeFLL to prior work on personalized FL, such as the baselines in [1,2,3,4].
>
> Moreover, we do not agree that showing superior performance on small models is irrelevant. In Federated Learning it is very common in practice to use small models due to the fact that we need to keep communication costs low and the client devices themselves are often too limited in terms of compute to be able to train huge models. For instance Google’s Gboard model, which they use in production, is a single layer LSTM, see [5,6].
>
> > The computational efficiency of the model is questionable given that the hypernetwork is at least a 100 times bigger than the client's model, and this is only the case for these particular experiments. It might require a wider net to generate larger personalized models. In other words, LeNet on MNIST and CIFAR-10/100 might only need 100 "ranks" to perform well, whereas on more complicated models and datasets, a wider net would be needed for good performance, which makes scalability an issue.
>
> If we understand you correctly, you are asking generally about the feasibility of using hypernetworks to generate large models. This is absolutely the case: there are a number of papers that use hypernetworks to generate very large models. For instance, [7] uses a hypernetwork to generate the parameters of a ResNet-50: “we can predict all 24 million parameters of ResNet-50 in less than a second either on a GPU or CPU” and [8] also generates ResNets using a hypernetwork. Note that -by design- in our method the hypernetwork is located on the server (as opposed to on client devices), so we are not so limited in terms of computational resources to run the hypernetwork.
>
> [1] A. Shamsian, A. Navon, E. Fetaya, and G. Chechik. Personalized federated learning using hypernetworks. ICML, 2021.
>
> [2] C. T. Dinh, N. H. Tran, and T. D. Nguyen. Personalized federated learning with moreau envelopes. NeurIPS, 2020.
>
> [3] L. Collins, H. Hassani, A. Mokhtari, and S. Shakkottai. Exploiting shared representations for personalized federated learning. ICML, 2021.
>
> [4] A. Fallah, A. Mokhtari, and A. E. Ozdaglar. Personalized federated learning with theoretical guarantees: A model-agnostic meta-learning approach. In NeurIPS, 2020.
>
> [5] A. Hard, K. Rao, R. Mathews, S. Ramaswamy, F. Beaufays, S. Augenstein, H. Eichner, C.  Kiddon, and D. Ramage. Federated learning for mobile keyboard prediction. arXiv:1811.03604, 2018.
>
> [6] X. Zheng, Y. Zhang, G. Andrew, C. A. Choquette-Choo, P. Kairouz, H. B. McMahan, J. Rosenstock, and Y. Zhang. "Federated Learning of Gboard Language Models with Differential Privacy." arXiv:2305.18465, 2023.
>
> [7] B. Knyazev, M. Drozdzal, G. W. Taylor, and R. Soriano, A.. Parameter prediction for unseen deep architectures. NeurIPS, 2021.
>
> [8]  D. Ha, A. M. Dai, and Q. V. Le. Hypernetworks. ICLR, 2017.

---

> ### Author Response · Authors · 2023-11-17
>
> > That is why I think experiments covering these settings are needed to conclude the practical benefits of PeFLL.
>
> We disagree that PeFLL should run with bigger target networks to be practical. As discussed above, small models are the default for personalized FL. For example, if in each round we had to send a ResNet50 with 24 million parameters to the clients, clients would suffer from large data transfer volumes already after a few rounds of participation.
>
>
> > Can you share some details about training time, memory consumption, and computational efficiency of PeFLL vs. regular models?
>
> Can you clarify this question? Are you referring to the overall runtime, memory and compute of the method in our simulations? Our method runs in around 7 hours, while baselines took between 5 and 10 hours to run. The overall memory consumption is approximately 1 GB. However, we do not think that these statistics are very reflective of what will happen in practice in FL given that runtime is most greatly influenced by how much compute a client device has, scheduling overhead, stragglers etc. During training the amount of compute that is run on each client is roughly the same for all methods, 50 local updates to the personalized model. PeFLL additionally has a single forward and backward pass on the (small) embedding network during training. When running on test clients PeFLL has much lower compute than all other personalized FL methods since it only requires a forward pass on the (small) embedding network while other methods need to run finetuning locally. PeFLL also has additional compute that happens on the server, but this is minimal and consists of a single forward and backward pass through the hypernetwork for each global training round. This is negligible given the large compute power of the server, and the fact that it is only a single pass.

---

> ### Comment · Reviewer_Couf · 2023-11-20
> **Reply to Authors' Comments by Reviewer**
>
> Thank you for addressing my comments.
>
> Note that I said “more modern” and not “larger”. Small models are indeed very relevant in FL. I totally agree with that, even though improvement in communication efficiency (either in terms of connection speed or algorithmic improvements) can allow the use for reasonably bigger models. However, I was actually more interested in seeing performance comparison on “modern” architectures regardless of size, particularly ones employing residual connections, attention, etc. This might be more relevant for cross-silo FL as models can in fact be a little big.
>
> Thanks for sharing some details about training.I understand from you that bigger models can be generated efficiently by hyper-networks, which is good. It would be interesting then to see how that holds up in your experiments. The server could indeed have great computational capabilities, but at least you should demonstrate that scalability would not be an issue (e.g. by showing that compute and/or memory scales linearly with model size, which is doable based on what you said).

---

> > ### Comment · Reviewer_Fw5z · 2023-11-23
> > **Some more points.**
> >
> > I agree that more modern networks would be interesting, however I believe that also for a more modern network and a given harsh computational resources constraint, a FedAvg global model will underperform compared to a personalized model through this method. Even though evidence for generalization across tasks & modalities is still somewhat lacking, I believe it a worthwhile method to include in ones personalization tool-box.
> > Another recent hypernetwork could be found here: https://hyperdreambooth.github.io/
> > While not predicting too many parameters (30k), the hypernetwork itself is a modern transformer architecture which scales quadratically with the number of network parameters.

---

### Author Response · Authors · 2023-11-23
**Summary of the main changes of the paper**

Dear Reviewers,


Thank you for your feedback and for the productive rebuttal period. We have uploaded a revised version of the manuscript to OpenReview with changes highlighted in blue. A summary of the main changes is as follows:


1) As suggested by the reviewers we have run additional experiments where a larger ResNet20 is generated by the hypernetwork as the personalized client model. Initial results of these experiments can be found in Appendix A.3, **ResNet Experiments**. Due to the short amount of time for Rebuttals (and the longer runtime of training ResNets) we did not had time to properly tune hyperparameters for these experiments. However, we do believe that given the reasonable performance these experiments do show the feasibility of generating such models as part of our pipeline.

For these experiments, we used the implementation of ResNet20 from https://github.com/chenyaofo/pytorch-cifar-models with the change that we removed the Batch Norm layers. The reason is that batch statistics (the running mean and the running var) would need to be computed at each client's data, and not by SGD or hypernetwork. Our framework could handle batch norm with outputting its learnable parameters and each client has to do some forward pass on a couple of batches to generate its running mean and running var. This needs changing the code of the clients and due to lack of time we ran the experiments on ResNet without batch norm (but with residual connections).


2) Also following the suggestions of reviewers we have run additional experiments on Shakespeare, which is another LEAF dataset and is not an image recognition task. The results for this can be found in Appendix A.3, **Shakespeare Dataset**


3) We have added an analysis of the performance of PeFLL as a function of communication cost. In particular we plot learning-curves with Accuracy vs. GB, and include a discussion of these curves. These can be found in Appendix A.3, **Comparison of communication and computation cost with FedAvg**. They show that for any fixed communication budget PeFLL substantially outperforms FedAvg.


4) Following the suggestions of Reviewer Fw5z we have removed label-masking from the main manuscript and updated the results in Table 1 to those of PeFLL without label masking. We include a discussion of label masking in the supplemental material, discussing it only as an extended scenario. See Appendix A.3, **Integrating known client label sets**.


5) Also following the suggestions of Reviewer Fw5z, we have run additional experiments for a challenging new personalization setting in which we assume that the Test Clients possess only an **unlabeled** training set. None of the personalized baselines we compare to are able to handle this scenario because they require labeled training data in order to produce a personalized model for a new client. PeFLL can solve this task by simply training the embedding network on the train clients without embedding the label information. See **Personalization for clients with only unlabeled data** in Section 4 of the main manuscript for full results and discussion.


Thanks again for all your feedback!

Best wishes,

The authors.

---

### Meta-Review · Area_Chair_oC57 · 2023-12-06

**Metareview:**

The authors agree that the paper addresses an interesting problem and that the solution inspired by learning-to-learn is sound.

There are outstanding concerns regarding how generalizable the approach is (beyond the experimental setup proposed in the original submission), experiments on more modern architectures, and how scalable the approach is wrt model size. There is strong reviewer support that while these issues, the benefits of the approach outweigh its negative points.

In particular, under the assumption that small, personalized models are preferred in FL to large unpersonalized settings (Rev Fw5z), the scalability issue is not critical.

The main outstanding concern is the limited experimental support for the approach. During the rebuttal, the authors did provide new experimental results, providing positive but mixed signals regarding the performance in a variety of settings.

I view this paper as borderline, and I recommend acceptance considering reviewer/author engagement in the discussions and reviewer support.

**Justification For Why Not Higher Score:**

There are outstanding concerns regarding how applicable the technical solution to a wide variety of real-world cases.

**Justification For Why Not Lower Score:**

The paper seems to contain interesting ideas and experimental results sufficient for the proof-of-concept

---

### Decision · Program_Chairs · 2024-01-16

Accept (poster)